# Genomic analysis of an ultrasmall freshwater green alga, *Medakamo hakoo*

Shoichi Kato [1], Osami Misumi[2], Shinichiro Maruyama [3,4,14], Hisayoshi Nozaki [5,6], Yayoi Tsujimoto-Inui[7], Mari Takusagawa[8], Shigekatsu Suzuki [6], Keiko Kuwata[9], Saki Noda[10], Nanami Ito[7], Yoji Okabe [7], Takuya Sakamoto[1], Fumi Yagisawa[11,12], Tomoko M. Matsunaga[7], Yoshikatsu Matsubayashi [10], Haruyo Yamaguchi[6], Masanobu Kawachi[6], Haruko Kuroiwa[13], Tsuneyoshi Kuroiwa [13✉] & Sachihiro Matsunaga [1,7✉]

Ultrasmall algae have attracted the attention of biologists investigating the basic mechanisms underlying living systems. Their potential as effective organisms for producing useful substances is also of interest in bioindustry. Although genomic information is indispensable for elucidating metabolism and promoting molecular breeding, many ultrasmall algae remain genetically uncharacterized. Here, we present the nuclear genome sequence of an ultrasmall green alga of freshwater habitats, *Medakamo hakoo*. Evolutionary analyses suggest that this species belongs to a new genus within the class Trebouxiophyceae. Sequencing analyses revealed that its genome, comprising 15.8 Mbp and 7629 genes, is among the smallest known genomes in the Viridiplantae. Its genome has relatively few genes associated with genetic information processing, basal transcription factors, and RNA transport. Comparative analyses revealed that 1263 orthogroups were shared among 15 ultrasmall algae from distinct phylogenetic lineages. The shared gene sets will enable identification of genes essential for algal metabolism and cellular functions.

[1] Department of Applied Biological Science, Faculty of Science and Technology, Tokyo University of Science, Noda, Chiba 278-8510, Japan. [2] Department of Biological Science and Chemistry, Faculty of Science, Graduate School of Medicine, Yamaguchi University, Yoshida, Yamaguchi 753-8512, Japan. [3] Department of Ecological Developmental Adaptability Life Sciences, Graduate School of Life Sciences, Tohoku University, Aobaku, Sendai 980-8578, Japan. [4] Graduate School of Humanities and Sciences, Ochanomizu University, Tokyo 112-8610, Japan. [5] Department of Biological Sciences, Graduate School of Science, The University of Tokyo, Hongo, Tokyo 113-0033, Japan. [6] Biodiversity Division, National Institute for Environmental Studies, Onogawa, Tsukuba, Ibaraki 305-8506, Japan. [7] Department of Integrated Biosciences, Graduate School of Frontier Sciences, The University of Tokyo, Kashiwa, Chiba 277-8562, Japan. [8] Department of Botany, Graduate School of Science, Kyoto University, Kyoto 606-8502, Japan. [9] Institute of Transformative Bio-Molecules (WPI-ITbM), Nagoya University, Chikusa, Nagoya 464-8602, Japan. [10] Division of Biological Science, Graduate School of Science, Nagoya University, Nagoya, Japan. [11] Center for Research Advancement and Collaboration, University of the Ryukyus, Okinawa 903-0213, Japan. [12] Graduate School of Engineering and Science, University of the Ryukyus, Okinawa 903-0213, Japan. [13] Department of Chemical and Biological Science, Faculty of Science, Japan Women's University, Tokyo 112-8681, Japan. [14]Present address: Department of Integrated Biosciences, Graduate School of Frontier Sciences, The University of Tokyo, Kashiwa, Chiba 277-8562, Japan. ✉email: tkuroiwa@fc.jwu.ac.jp; sachi@edu.k.u-tokyo.ac.jp

Microalgae are microscopic unicellular phytoplankton found in freshwater, seawater, and sediment, and are invisible to the naked eye[1]. Microalgae form the basis of the food chain in aquatic ecosystems, and play important roles in carbon dioxide capture and sequestration through photosynthesis[2]. Despite their ecological importance in providing energy to support all higher trophic levels, more than 70% of the species are estimated to remain unidentified[3]. Microalgae have been used to produce highly functional foods, biofuels, and materials used in cosmetics[1]. To improve the production efficiency and profitability of current algal culture systems, demand is increasing for especially small microalgae that can be cultured at high densities.

We focused on *Medakamo hakoo* (Chlorophyta), an ultrasmall algal species found in freshwater that potentially may provide notable insights into genome biology of algae. *Medakamo hakoo* was first identified and reported in 2015[4]. A previous study involving DNA staining revealed that *M. hakoo* likely has the smallest known nucleus among Archaeplastida species[5]. Although some microalgae inhabiting seawater and hot springs have extremely simple genomes[6–10], relatively few freshwater algae with extremely small genomes have been reported. Genomic analysis of *M. hakoo* is expected to produce useful information for future investigations on effective culture methods for optimal production of useful substances. Genomic information for *M. hakoo* will also contribute to understanding how eukaryotic phototrophs thrive in diverse environments. In addition, comparison of the genomes of *M. hakoo* and other ultrasmall algal species is an effective strategy for identifying the gene set common to algal species and genes common to green algal lineages.

In this study, we first characterized the morphological features and synchronization of the cell cycle of *M. hakoo*. Next, the *M. hakoo* genome sequence was assembled from long reads generated using the PacBio RSII system in conjunction with RNA-seq analysis of the transcriptome. Finally, comparison of the genomes of *M. hakoo* and 14 other microalgal species revealed that *M. hakoo* has one of the smallest genomes among freshwater algae, and 1263 gene families conserved among microalgae were identified.

## Results

**Investigation of *M. hakoo* cellular characteristics**. To characterize *M. hakoo* morphology, we first used SYBR Green I stain to label *M. hakoo*, *Cyanidioschyzon merolae*, and *Saccharomyces cerevisiae* nuclei, and observed that the fluorescence intensity of the *M. hakoo* nuclei was similar to that of *C. merolae* and *S. cerevisiae* nuclei (Fig. 1a, b, Supplementary Fig. 1). *Cyanidioschyzon merolae* is an ultrasmall unicellular red alga with the smallest genome in Rhodophyta[6,7]. Fluorescence and transmission electron microscopic examination indicated that *M. hakoo* cells were approximately 1 µm in diameter and contained relatively few organelles, with only a single mitochondrion and chloroplast (Fig. 1c–g, Supplementary Fig. 2). Notably, a specific electron-dense structure in the nuclear peripheral region and thick cell walls were typical characteristics of *M. hakoo* cells (Fig. 1f, Supplementary Fig. 2d, e). Another structural feature observed in *M. hakoo* cells was the accumulation of starch aggregates in the chloroplast (Supplementary Fig. 2d). In *C. merolae*, starch aggregated in the cytoplasm (Supplementary Fig. 2a, c). Additionally, phycobilisomes were undetectable in *M. hakoo* chloroplasts (Fig. 1f, Supplementary Fig. 2). To examine the *M. hakoo* cell division pattern, we cultured cells under a light–dark cycle to obtain highly synchronized cells, and detected the following cell-cycle stages: a single-cell stage (I), a two-cells-combined stage (II), a tetrad stage (III), and a dissection stage

(IV) (Fig. 1h, i, Supplementary Fig. 3). In addition, *M. hakoo* cells cultured in nitrogen-depleted medium typically formed lipid droplets, similar to the response of the oil-rich alga *Botryococcus braunii* (Supplementary Fig. 4)[11–13].

**Sequencing and evolutionary analysis of the *M. hakoo* genome**. From the long-read sequencing analysis, we obtained 18 contigs via a de novo sequence assembly (Table 1, Supplementary Table 1), of which two contigs were annotated as organellar genomes because they were circular sequences. To perform a phylogenetic analysis, we used the *M. hakoo* organellar genome, which we previously described[14]. A phylogenetic tree was constructed on the basis of plastid genome sequences from 62 chlorophyte taxa using the maximum-likelihood (ML) method (Fig. 2a). The resulting tree suggested that *M. hakoo* is classifiable in the class Trebouxiophyceae. Additionally, *M. hakoo* is likely evolutionarily related to *B. braunii*, which shows potential for algal fuel production[15,16] (Supplementary Fig. 4).

In our phylogenetic analysis, *Medakamo* and *Choricystis* formed a small clade sister to *Botryococcus* (Fig. 2a). Many algal strains originating from various freshwater habitats were recently identified as *Choricystis* species, and their *rbcL* sequences are available in the NCBI database (e.g., Novis et al.[17]). In addition, three *Choricystis* species were identified mainly on the basis of phylogenetic analyses by Pröschold and Darienko[18]. To more precisely resolve the phylogenetic relationships between *Medakamo* and *Choricystis*, 54 *Choricystis rbcL* sequences in the NCBI database, two new *rbcL* sequences from strains studied by Pröschold and Darienko[18], and the *Medakamo rbcL* sequence were included in a phylogenetic analysis, with *Botryococcus* sequences as the outgroup (Fig. 2b). The phylogenetic tree robustly resolved two sister clades (with bootstrap values of 83% or higher) that corresponded to *Choricystis* and the new genus *Medakamo*. *Medakamo* comprises *M. hakoo* sp. nov. and *M. limnetica* comb. nov. (= *Choricystis limnetica*) and can be clearly distinguished from *Choricystis* and other freshwater green algae on the basis of their phylogenetic position and differences in cell morphology (Supplementary Note 1). Thus, *Medakamo* is recognized as a new genus within the class Trebouxiophyceae.

**Nuclear genome analysis**. The 16 non-organellar contigs were chromosomal sequences flanked by telomere sequences (5′-TTAGGG-3′). The total contig size was 15.8 Mb (Tables 1, 2), which was larger than the genome size estimated by fluorescence microscopy[5]. Thus, the contigs represented almost the complete genome sequence.

Next, we validated the genome assembly by confirming several benchmarks. The sequence coverage of the assembly was 246.8×. To further evaluate the assembled genome, we conducted a Benchmarking Universal Single-Copy Orthologs (BUSCO) analysis[19,20]. After removing organellar contigs, 89.5% of the assembly comprised complete BUSCOs (Supplementary Fig. 5). We also mapped RNA-seq reads to the contigs, with 98.6% of the reads successfully aligned. Moreover, tRNAs corresponding to all 20 amino acids were identified (Supplementary Table 2).

Using the BRAKER2 software[21], we identified 7629 candidate protein-coding sequence (CDS) regions in the nucleus, and annotated these sequences using eggNOG-mapper[22] (Supplementary Data 1) and GhostKOALA[23] (Supplementary Data 2). Amino acid sequences were obtained for 91.0% of the complete BUSCOs (Supplementary Fig. 5).

**Effect of a high G + C content on the amino acid content in the *M. hakoo* proteome and gene expression**. A high G + C content (73 mol%) is one characteristic of the *M. hakoo* genome (Table 2).

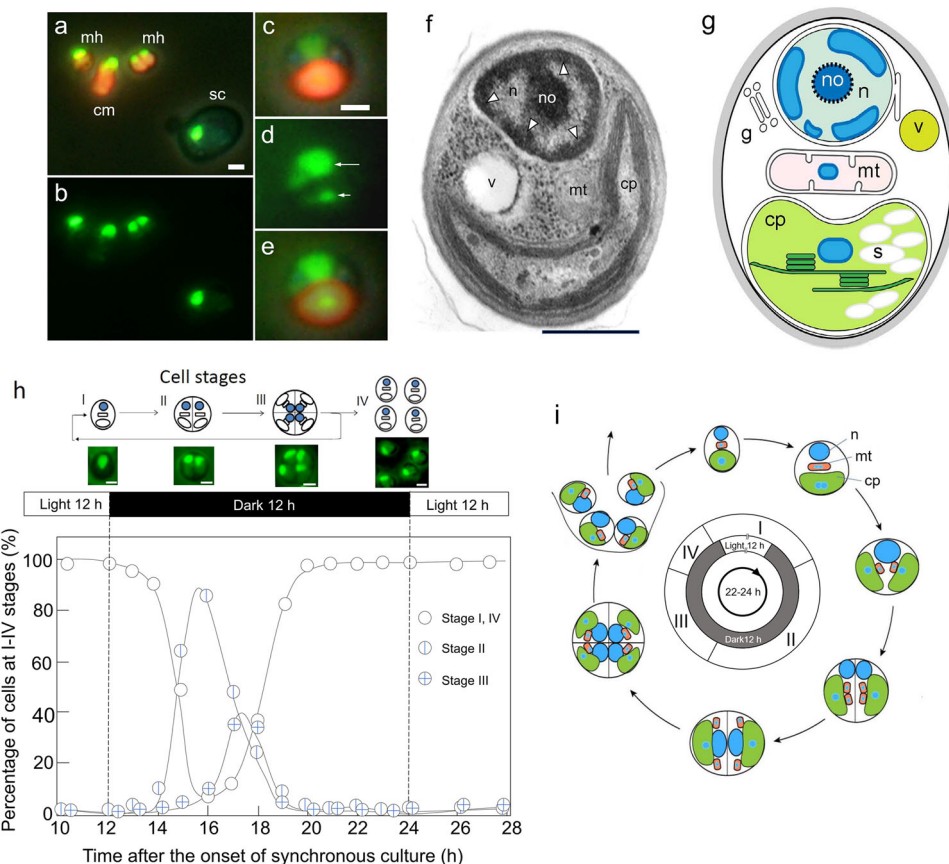

**Fig. 1 Morphological characteristics of *Medakamo hakoo*. a** Fluorescence images merged with a phase-contrast image of *M. hakoo* (mh), *Cyanidioschyzon merolae* (cm), and *Saccharomyces cerevisiae* (sc). SYBR Green signals appear green. Red signals are autofluorescence from the chlorophyll in chloroplasts. The scale bar indicates 1 μm. **b** Image of the SYBR Green fluorescence only for the sample presented in panel **a**. **c–e** Highly magnified images of *M. hakoo*. **c** Phase-contrast and autofluorescence image. The scale bar indicates 1 μm. **d** SYBR Green signals. Long and short arrows indicate the nuclear and chloroplast DNA, respectively. **e** Merged image of (**c**) and (**d**). **f** Transmission electron microscopy image of *M. hakoo*; no, n, v, mt and cp represent the nucleolus, nucleus, vacuole, mitochondrion, and chloroplast, respectively. Arrowheads indicate electron-dense structures. The scale bar indicates 500 nm. **g** Schematic image of the *M. hakoo* cell structure; no, n, g, mt, v, s, and cp indicate the nucleolus, nucleus, Golgi apparatus, mitochondrion, vacuole, starch, and chloroplast, respectively. **h** Synchronization culture of *M. hakoo*. The schematic diagram with images of SYBR Green fluorescence presents the following cell stages: single-cell stage (I), two-cells-combined stage (II), tetrad stage (III), and dissection stage (IV). The *M. hakoo* cells were treated with a 12-h light/12-h dark cycle. Totally, 4268 cells were counted. The counts at each time point are provided in Supplementary Data 6. The scale bars indicate 500 nm. **i** Schematic diagram of the *M. hakoo* lifecycle; n, mt, and cp indicate the nucleus, mitochondrion, and chloroplast, respectively.

| Table 1 Basic data for the *Medakamo hakoo* genome. | |
| --- | --- |
| Total contig number | 18 |
| Organella derived contig number | 2 |
| Total contig (bp) | 15,811,321 |
| GC (bp) | 11,495,275 |
| GC contents (%) | 72.7 |
| Average contig length (bp) | 878,407 |
| Maximum contig length (bp) | 1,473,234 |
| N50 contig length (bp) | 1,246,908 |
| total subreads length (bp) | 3,902,249,383 |
| Coverage | 246.8 |
| Predicted nuclear CDS number | 7629 |
| Predicted plastid CDS number | 152 |
| Predicted mitochondrial CDS number | 76 |

The G + C content was high in protein-coding regions as well (Supplementary Table 3). A high G + C content in the protein-coding regions may increase the number of certain types of amino acids in the resultant proteins (Fig. 3a). We analyzed the amino acid composition of the predicted protein sequences in *M. hakoo* and compared them with the amino acid composition in other microalgae. Our analysis demonstrated that *M. hakoo* proteins contained many alanine, glycine, and proline residues. More specifically, alanine was more abundant in the *M. hakoo* proteome than in the *C. merolae* proteome (Fig. 3b). To investigate the relationship between gene expression and the G + C content in the CDSs, we plotted the G + C content and the transcripts per million (TPM) value for each gene (Fig. 3c), which revealed a negative correlation between these two factors. A negative correlation was also detected between the alanine content and the TPM value for each gene (Fig. 3d). These results suggested that highly expressed genes (e.g., housekeeping genes) were relatively unaffected by the bias toward a high G + C content in the *M. hakoo* genome.

We further analyzed *M. hakoo* expression patterns using proteomics, which detected more than 3000 unique peptides across samples (Supplementary Data 3). All codons were represented in the proteins detected (Table 3, Supplementary Fig. 6), suggesting that *M. hakoo* cells use a standard codon table, with the caveat that mass spectrometry cannot distinguish between leucine and isoleucine residues.

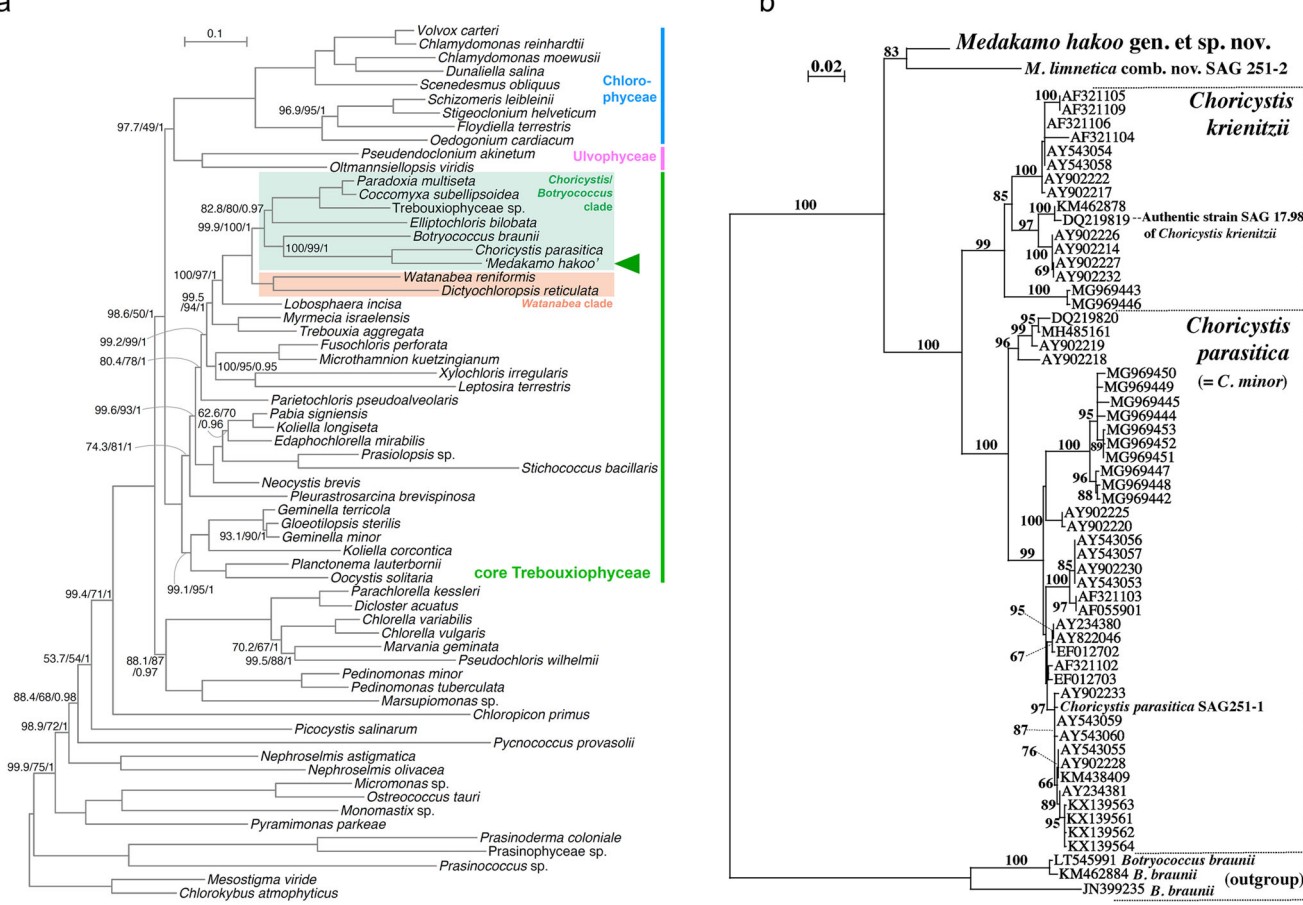

**Fig. 2 Evolutionary analyses of the *Medakamo hakoo* genome. a** Phylogeny of 62 chlorophytes inferred using a dataset comprising 15,561 amino acid positions in 79 cpDNA-encoded proteins. The best maximum-likelihood (ML) tree is presented, with support values at the nodes (the approximate likelihood-ratio test support/non-parametric bootstrap support values analyzed by IQ-TREE/posterior probability calculated using PhyloBayes). A lack of values at branches reflects 100% support from all analyses. **b** Phylogenetic relationships between *M. hakoo* and 54 algal strains identified as *Choricystis* species in the NCBI database (https://www.ncbi.nlm.nih.gov/), *Choricystis parasitica* SAG 251-1 (NIES-1436), and *Medakamo limnetica* comb. nov. SAG 251-2 [= *Choricystis limnetica*[18], according to the ML analysis of the chloroplast Rubisco large subunit gene (*rbcL*)]. Bootstrap values (50% or higher) for 1000 replications are presented at the branches. Branch lengths are proportional to the evolutionary distances indicated by the scale bar above the tree.

**Table 2 Comparison of genome data for five microalgal species.**

|                      | *C. merolae* | *M. hakoo* | *O. tauri* | *C. variabilis* | *C. reinhardtii* |
|----------------------|--------------|------------|------------|-----------------|------------------|
| Genome size (Mb)     | 16.5         | 15.8       | 12.6       | 46.2            | 121              |
| No. of chromosomes   | 20           | 16         | 20         | 12              | 17               |
| No. of total genes   | 5010         | 7857       | 7765       | 9892            | 14,415           |
| GC content (%)       | 55           | 73         | 58         | 67              | 64               |

**Relationship between the extremely high G + C content of the *M. hakoo* genome and the notable accumulation of the guanine quadruplex consensus sequence**. To determine the reason for the high G + C content in the *M. hakoo* genome, we examined the G + C content-related characteristics of the genome sequence. We focused on the guanine quadruplex (G4) structure, which is a non-B DNA structure that forms in guanine-abundant DNA regions[24–26]. The guanine nucleotides form a tetrad structure that is involved in various biological processes, including DNA replication, transcription, meiotic double-strand break, and telomere maintenance[24–26]. To predict the abundance of G4 in the genome of *M. hakoo* and other organisms, we used pqsfinder software, which can detect the G4 consensus sequence in DNA sequences[27] (Fig. 3e). The frequency of the predicted G4 regions was highest in *M. hakoo* among the analyzed organisms. Notably,

the frequency was higher than that in *Streptomyces coelicolor*, which has a genome with one of the highest G + C contents among prokaryotes.

**Comparison between the *M. hakoo* genome and the genomes of other species**. We compared the *M. hakoo* genome with previously sequenced genomes in terms of their size and gene number (Fig. 4a; Table 2). The genome of *M. hakoo* was one of the smallest among Viridiplantae species and was smaller than that of *Auxenochlorella protothecoides*, which has the smallest genome previously known among freshwater green algae (Fig. 4a). We classified the *M. hakoo* nuclear genes according to the Kyoto Encyclopedia of Genes and Genomes (KEGG) Orthology[28] and determined the number of genes belonging to each functional group. First, we calculated the number of genes in

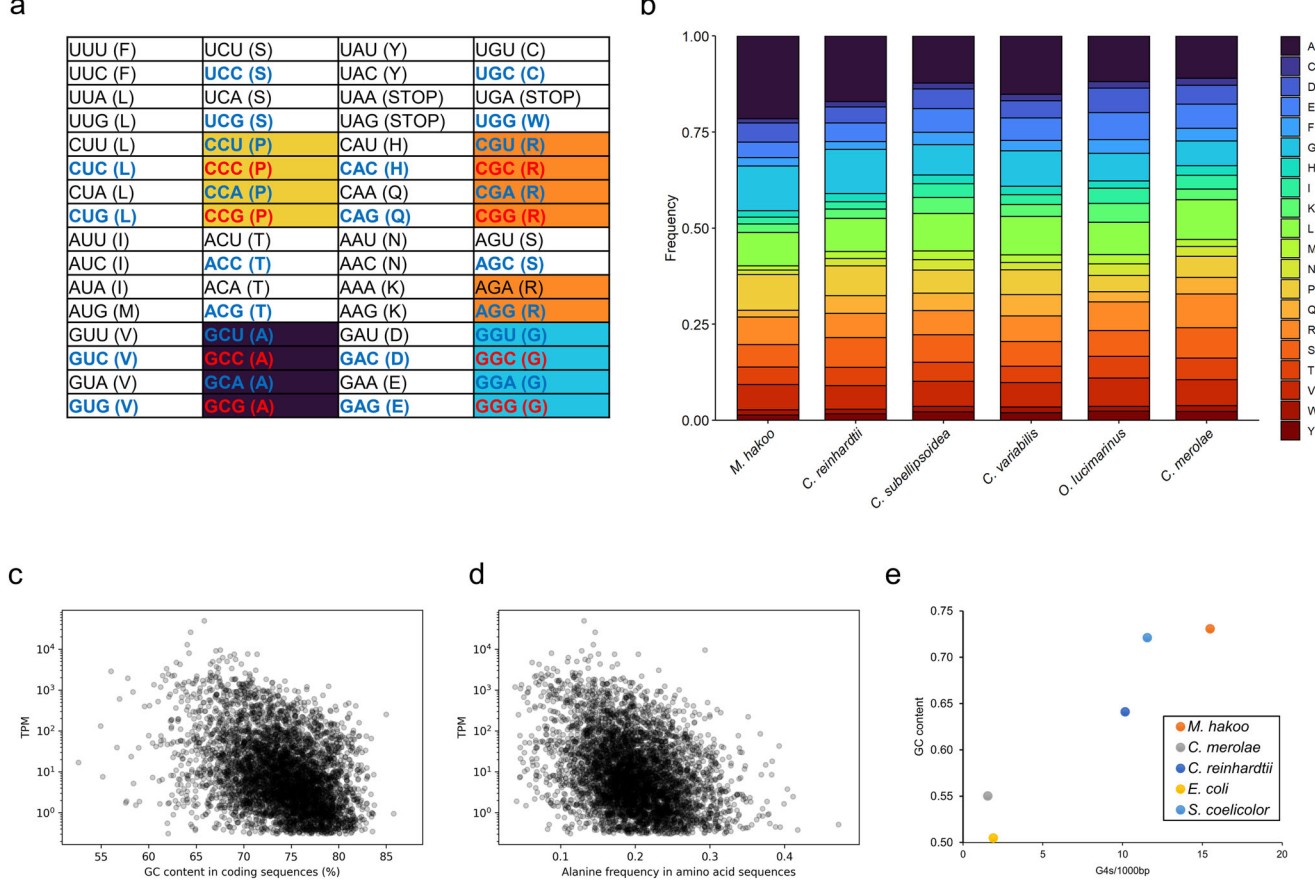

**Fig. 3 Effect of a high genomic G + C content. a** Codon table. Codons including two G or C are highlighted in blue, whereas those including three G or C are highlighted in red. Alphabets in parentheses indicate single letter designations of amino acids. STOP represents a termination codon. **b** Amino acid composition in the predicted proteomes of *Medakamo hakoo* and other microalgae. **c** Relationship between gene expression and the G + C content of each coding sequence. **d** Relationship between gene expression and the frequency of alanine in each protein. **e** Guanine quadruplexes predicted using the pqsfinder software.

the most enriched KEGG Orthology categories and compared the results with those for other green algae (Fig. 4b). Overall, *M. hakoo* had fewer genes than the other examined species and, most notably, fewer genes belonging to the 'Genetic information processing' category. Next, we determined the number of genes annotated with specific pathways categorized as 'Genetic information processing'. For many of these pathways, *M. hakoo* tended to have fewer genes compared with other green algal species (Fig. 4c). On the basis of the gene sets, the pathways associated with 'Basal transcription factors' and 'Nucleocytoplasmic transport' were considerably simpler in *M. hakoo* than in *Ostreococcus lucimarinus*, a marine green alga with a smaller genome than *M. hakoo* (Fig. 4c).

**Conservation of important biological pathways**. To more precisely determine the characteristics of the *M. hakoo* genome, we analyzed the conservation of genes associated with major biological pathways. Although almost all fundamental photosynthesis-related genes were conserved in *M. hakoo*, genes encoding stress-related light-harvesting complex (LHC)-like proteins (LHCSR) were not detected in the *M. hakoo* genome (Supplementary Figs. 7, 8). Regarding the photoreceptors, *M. hakoo* had cryptochrome and phototropin orthologs, but lacked a phytochrome homolog. Although several green algae have phytochromes[29], their absence in *M. hakoo* is consistent with the lack of phytochromes in the green alga *Chlamydomonas reinhardtii*[30] and the red alga *Cyanidioschyzon merolae*[6].

Our analysis also confirmed that *M. hakoo* had a set of conserved canonical histones, including H2A, H2B, H3, and H4. However, we did not detect an ortholog of the linker histone H1 encoded in the *M. hakoo* genome. To confirm the conservation of other chromosomal proteins, we analyzed structural maintenance of chromosomes (SMC) protein complexes (cohesin, condensin I/II, and SMC5/6), which are key regulators of chromosomal organization, dynamics, and stability[31]. Our analysis indicated that canonical components of all SMC complexes were conserved in *M. hakoo* (Supplementary Table 4).

Notably, relatively few nuclear envelope (NE)-related genes were identified in the *M. hakoo* genome. We investigated whether the NE-related genes were conserved in *M. hakoo*. The NE is composed of the outer nuclear membrane, the inner nuclear membrane, and the nuclear pore complex, which is mechanically supported by the nuclear lamina beneath the inner nuclear membrane[32,33]. We determined that the nuclear pore complex components (nucleoporins) were highly conserved, in contrast to other NE-related factors, which were minimally conserved. Only one ortholog of the mid-Sad1/UNC84 domain-containing protein (mid-SUN)[34] was a conserved inner nuclear membrane protein, and none of the outer nuclear membrane and nuclear lamina proteins were conserved.

RNA interference (RNAi) is the molecular mechanism that regulates gene expression via small RNA and related proteins. Although RNAi-related proteins, such as Dicer and Argonaute, are widely conserved among eukaryotic organisms[35,36], genes

**Table 3 Codon counts in the *Medakamo hakoo* genome regions for the peptides detected during the proteomic analysis.**

| Codon (amino acid) | Experiment 1 (PSMs>1) | Experiment 2 (PSMs>1) |
|---|---|---|
| AAA (K) | 6309 | 5194 |
| AAC (N) | 12,898 | 10,948 |
| AAG (K) | 18,094 | 15,609 |
| AAU (N) | 997 | 822 |
| ACA (T) | 1786 | 1371 |
| ACC (T) | 14,419 | 12,212 |
| ACG (T) | 21,878 | 18,633 |
| ACU (T) | 1130 | 921 |
| AGA (R) | 379 | 293 |
| AGC (S) | 5084 | 4266 |
| AGG (R) | 1615 | 1282 |
| AGU (S) | 1098 | 950 |
| AUA (I) | 82 | 61 |
| AUC (I) | 18,464 | 15,502 |
| AUG (M) | 10,577 | 8950 |
| AUU (I) | 2817 | 2417 |
| CAA (Q) | 3073 | 2532 |
| CAC (H) | 14,029 | 11,542 |
| CAG (Q) | 13,184 | 11,308 |
| CAU (H) | 608 | 477 |
| CCA (P) | 3572 | 2811 |
| CCC (P) | 24,383 | 20,823 |
| CCG (P) | 31,004 | 26,653 |
| CCU (P) | 3661 | 2873 |
| CGA (R) | 983 | 725 |
| CGC (R) | 30,194 | 25,637 |
| CGG (R) | 15,663 | 13,210 |
| CGU (R) | 2280 | 1860 |
| CUA (L) | 464 | 378 |
| CUC (L) | 15,922 | 13,591 |
| CUG (L) | 46,175 | 39,648 |
| CUU (L) | 811 | 667 |
| GAA (E) | 11,179 | 9282 |
| GAC (D) | 38,955 | 33,385 |
| GAG (E) | 23,857 | 20,700 |
| GAU (D) | 2241 | 1886 |
| GCA (A) | 3850 | 3068 |
| GCC (A) | 69,968 | 59,544 |
| GCG (A) | 66,218 | 56,989 |
| GCU (A) | 5892 | 5020 |
| GGA (G) | 2275 | 1941 |
| GGC (G) | 63,865 | 54,584 |
| GGG (G) | 14,858 | 12,814 |
| GGU (G) | 4532 | 3960 |
| GUA (V) | 245 | 187 |
| GUC (V) | 32,137 | 27,565 |
| GUG (V) | 21,814 | 18,532 |
| GUU (V) | 1116 | 957 |
| UAA (STOP) | 80 | 59 |
| UAC (Y) | 13,834 | 11,742 |
| UAG (STOP) | 97 | 101 |
| UAU (Y) | 612 | 487 |
| UCA (S) | 1326 | 1041 |
| UCC (S) | 22,010 | 18,535 |
| UCG (S) | 13,880 | 11,897 |
| UCU (S) | 2287 | 1972 |
| UGA (STOP) | 864 | 782 |
| UGC (C) | 8086 | 6759 |
| UGG (W) | 9595 | 8077 |
| UGU (C) | 963 | 767 |
| UUA (L) | 358 | 313 |
| UUC (F) | 14,709 | 12,472 |
| UUG (L) | 4847 | 4122 |
| UUU (F) | 7298 | 6096 |

Experiments 1 and 2 were technical replicates. STOP represents a termination codon.

degradation of its components. However, no orthologs of autophagy-related genes are known in red algae[38]. By screening the *M. hakoo* genome, we identified conserved autophagy-related (ATG) genes (Supplementary Table 6), including some that encode core ATG proteins required for autophagosome formation[39]. This indicates that the autophagy system is conserved in chlorophytes, but not in rhodophytes.

**Analysis of orthogroup composition in *M. hakoo* and other algae.** For further analysis of the gene composition in *M. hakoo* and other algae, we analyzed orthogroups of *M. hakoo* and 14 other algal species comprising 11 green algae (*A. protothecoides, Bathycoccus prasinos, Chlamydomonas reinhardtii, Chlorella variabilis, Coccomyxa subellipsoidea, Micromonas commoda, Micromonas pusilla, Monoraphidium neglectum, O. lucimarinus, O. tauri,* and *Volvox carteri*) and three red algae (*Chondrus crispus, Cyanidioschyzon merolae,* and *Galdieria sulphuraria*) with OrthoFinder[40,41]. The composition of each algal orthogroup is summarized in Supplementary Data 4 and the data were analyzed using principal component analysis (Fig. 5a). The microalgal genomes were classifiable into several groups according to their orthogroup composition. *Medakamo hakoo* belonged to a group composed of trebouxiophycean algae (*A. protothecoides, C. subellipsoidea,* and *C. variabilis*) and had the smallest genome in this group.

**Analysis of shared genes among microalgae.** Given that *M. hakoo* had the smallest genome known among trebouxiophycean microalgae, we compared the orthogroups of *M. hakoo* and 14 microalgal species from other lineages to identify the gene set common to these microalgae (C15 gene set). We added *S. cerevisiae* gene sets to the OrthoFinder analysis to divide the C15 gene set into the following two classes: genes typically conserved in eukaryotes (CE) gene set and algae-specific (AS) genes. The AS gene set was defined with a purposefully lenient criterion to maximally capture the potential diversity of microalgal orthologs.

The 1263 orthogroups were classified into the C15 gene set, of which 984 and 279 orthogroups comprised the CE and AS gene sets, respectively (Fig. 5b, Supplementary Data 5). The CE and AS gene sets in the *M. hakoo* genome tended to be more highly expressed than the genes not included in these sets (Fig. 5c, Supplementary Table 7), indicating that the gene sets include many housekeeping genes. Next, we compared each gene set with the *M. hakoo* genome in terms of the proportion of genes assigned to particular KEGG Orthology categories, which revealed that the proportion of genes associated with metabolism was substantially higher in the AS gene set (Fig. 5d).

We also analyzed the enriched pathways in these gene sets. To evaluate the extent of the enrichment, we mapped the C15 and AS gene sets to KEGG pathways and calculated the ratio of the mapped genes in each pathway. We considered that a high ratio of AS genes to C15 genes indicated that the pathway strongly contributed to the characteristics of the AS gene set. The highly enriched pathways were associated with metabolism and photosynthesis (Fig. 5e, Table 4). Notably, the AS genes were mainly associated with pathways involved in secondary metabolism (Fig. 5e, Table 4).

**Discussion**
In this study, we revealed that the freshwater green alga *M. hakoo* is an ultrasmall microalgae with cells 1 µm in diameter that can accumulate useful substances, including starch and lipids, and thus may be of utility for bioproduction. The most unique characteristic of *M. hakoo* is strong synchronization of the cell cycle under a light–dark cycle. Recently, it was shown that the key

encoding these proteins were not detected in the *M. hakoo* genome. The genomes of microalgae, including *C. merolae* and *O. tauri*, also appear to have lost RNAi-related genes (Supplementary Table 5).

Autophagy involves the transport of a cytoplasmic cargo to the lysosome for degradation[37]. This mechanism is crucial for cell homeostasis because it allows the cell to recycle itself via the

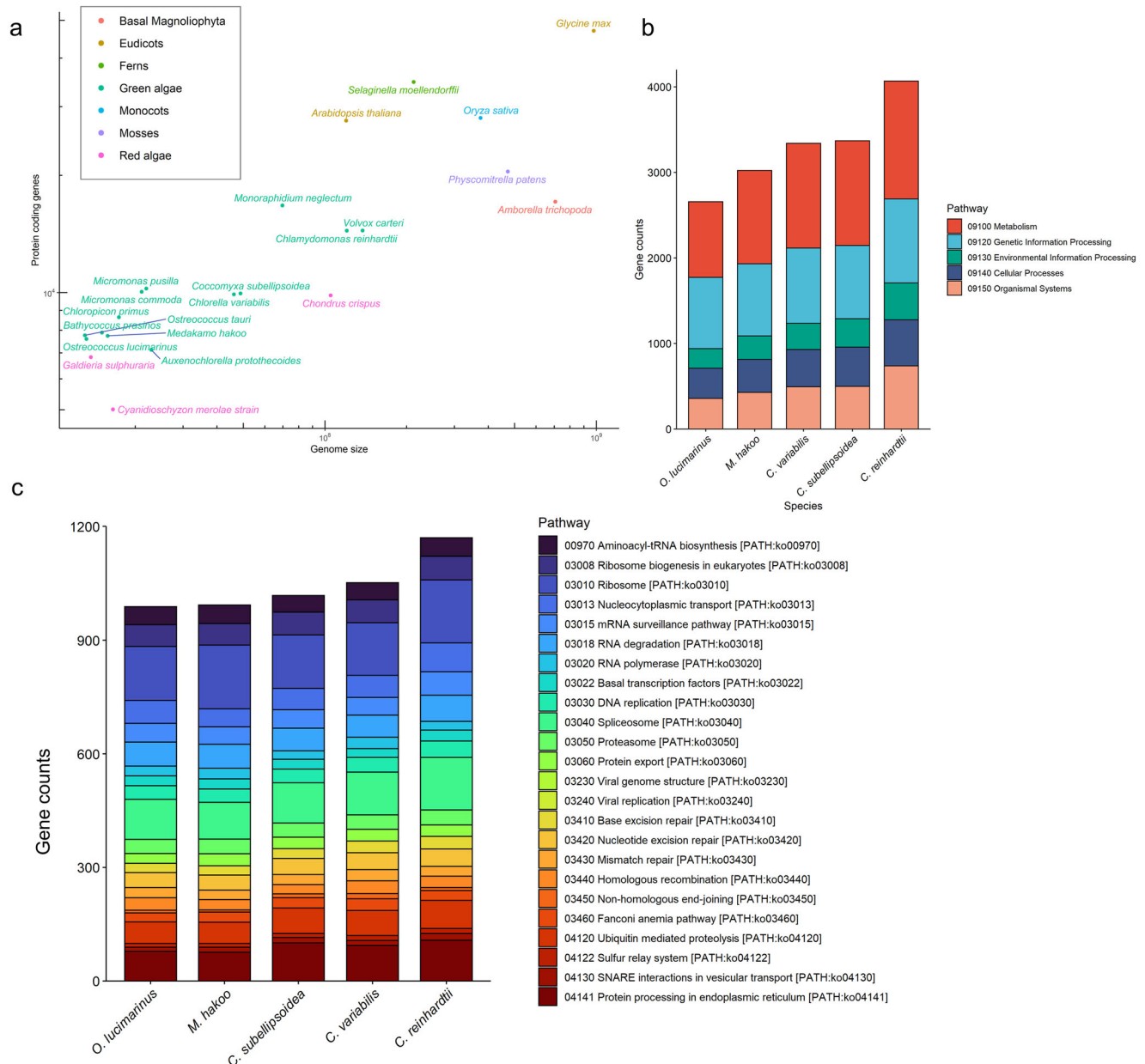

**Fig. 4 Gene number and composition of the *Medakamo hakoo* genome. a** Plot of the protein-coding genes and the total genome size of plants. **b** Number of green algae genes belonging to each KEGG Orthology category. The first-level classification was used for counting genes. **c** Number of green algae genes belonging to the 'Genetic information processing' KEGG Orthology category. The third-level classification was used for counting genes.

to maintaining dense algal cultures is to avoid clogging of the photosynthetic electron flow by appropriate regulation of the timing of light–dark cycles[42]. Thus, this attribute confers the potential to contribute to effective and stable bioproduction of useful substances with uniform quality[1]. In addition to these characteristics, *M. hakoo* has an extremely small genome. On the basis of the available information on Viridiplantae species in the KEGG Organisms database[43] and the JGI genome portal[44,45], *M. hakoo* likely possesses one of the smallest genomes among freshwater Viridiplantae species. This finding suggests that maintaining a small genome and small cells is advantageous for survival, not only in seawater and extreme environments, but also in common freshwater environments. According to the package effect[11–13], small cells are advantageous for photosynthesis. Although the mechanisms underlying the maintenance of ultra-small cells remain unknown in microalgae, generic factors, such

as photosynthetic efficiency, may have induced a decrease in genome size or suppressed genomic expansion.

In green plants, LHCSR and PSBS are conserved LHC-like proteins with functions associated with non-photochemical quenching[46]. Although there is some uncertainty regarding the underlying molecular mechanism, LHCSR plays a major role in the dissipation of excess light energy in *C. reinhardtii*. Regarding land plants, both LHCSR and PSBS help to dissipate excess energy in the moss *Physcomitrium*, but *Arabidopsis thaliana* lacks LHCSR and uses PSBS as a central component of its light energy dissipation machinery[46]. Additionally, PSBS is universally conserved in Viridiplantae, with the exception of bryopsidalean algae (*Ostreobium* and *Caulerpa*), which may have adapted to diverse light conditions[47]. In *M. hakoo*, the absence of LHCSR and the presence of the plastid genome-encoded cemA, which is required for the tolerance of *Chlamydomonas* to high-light

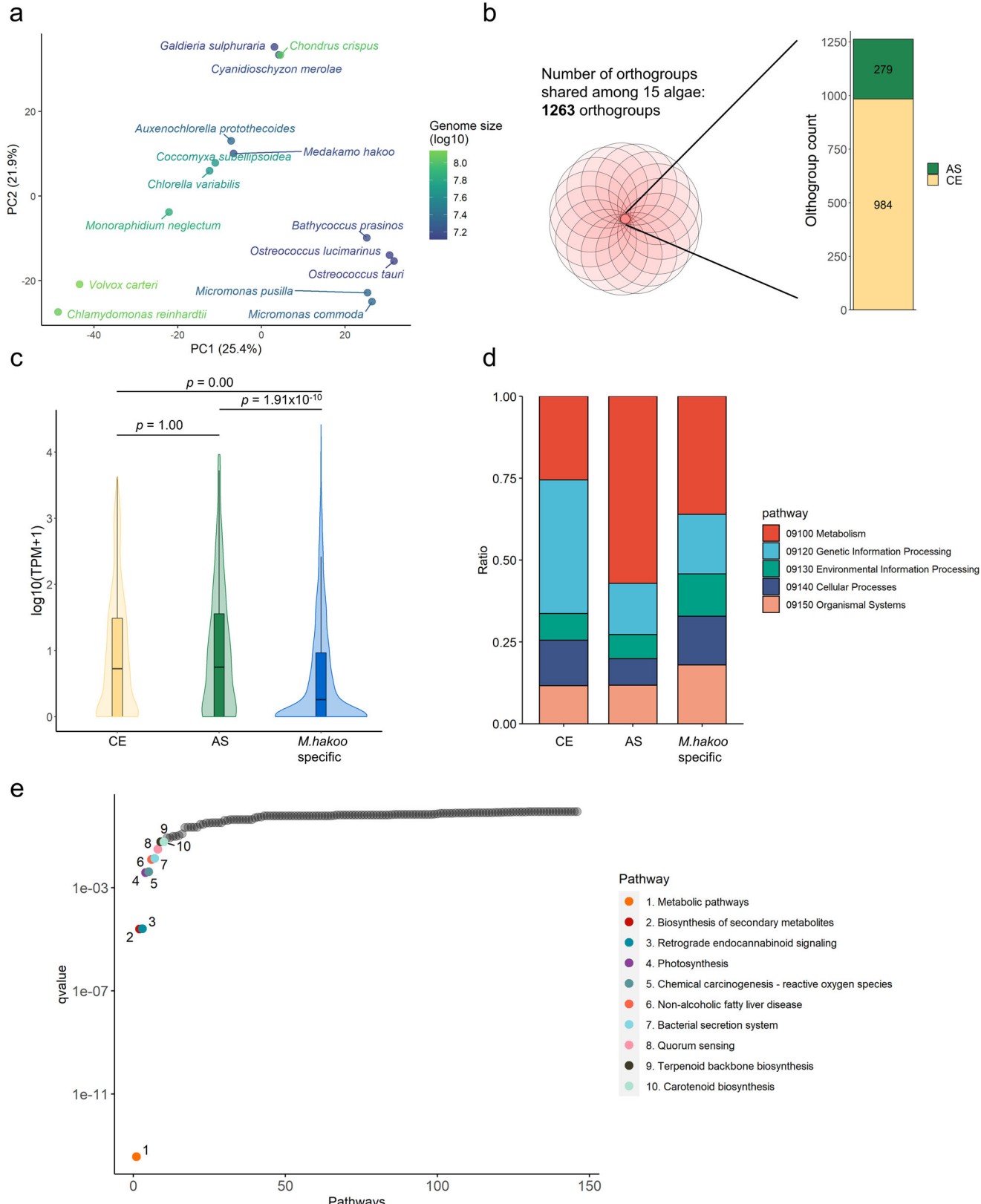

stress[48], suggest that this alga may have a unique high-light acclimation mechanism that may have convergently evolved in flowering plants and in an early-branching green alga (*Chloropicon*) (Supplementary Figs. 7, 8).

The *M. hakoo* genome does not contain a histone H1 gene CDS. Histone H1 includes the H15 domain, which is a globular domain comprising a winged-helix motif[49]. Green algae, including *Chlorella sorokiniana*[50] and *Haematococcus lacustris*[51], and red algae, including *C. merolae*[6], have histone H1 orthologs or H15 domain-containing proteins. Similar to *M. hakoo*, the genome of *O. tauri* lacks a gene encoding a protein with the H15 domain[8]. Because the genes encoding SMC proteins, including

**Fig. 5 Analysis of orthogroups among *Medakamo hakoo* and other microalgae. a** Principal component analysis of the orthogroup composition of algal genomes. Matrix data from Supplementary Data 4 was analyzed. The percentages represent proportion of variance of each axis. **b** Shared orthogroups among microalgae. Fifteen algae shown in Fig. 5a shared 1263 orthogroups, which were classified into the gene set conserved in eukaryotes (CE) and the algae-specific (AS) gene set. **c** Gene expression of CE, AS, and the *M. hakoo*-specific gene sets. The edges of the boxplot indicate the first and third quartiles, and the center line indicates the median. The number of genes in the CE, AS and the *M. hakoo*-specific gene sets used in this analysis was 984, 279 and 2707, respectively. Brunner-Munzel test[89] with Bonferroni correction was used for comparison among gene sets. **d** Gene ontology analysis of the complete genome and CE and AS gene sets. **e** Pathway enrichment analysis of the AS gene set. The CE and AS gene sets were mapped to KEGG pathways. The mapped ratio of AS to CE genes was calculated for each pathway. Significantly enriched pathways ($q$-value < 0.05) are presented in different colors.

**Table 4 Enriched KEGG Orthology categories in the algae-specific gene set.**

| KEGG_ID | Description | GeneRatio | BgRatio | $p$-value | p.adjust | $q$-value |
|---|---|---|---|---|---|---|
| ko01100 | Metabolic pathways | 88/123 | 283/737 | 2.84E−16 | 4.15E−14 | 3.68E−14 |
| ko01110 | Biosynthesis of secondary metabolites | 50/123 | 165/737 | 3.83E−07 | 2.80E−05 | 2.48E−05 |
| ko04723 | Retrograde endocannabinoid signaling | 10/123 | 12/737 | 5.96E−07 | 2.90E−05 | 2.57E−05 |
| ko00195 | Photosynthesis | 6/123 | 7/737 | 0.000118 | 0.004296 | 0.003809 |
| ko05208 | Chemical carcinogenesis - reactive oxygen species | 10/123 | 18/737 | 0.000159 | 0.004629 | 0.004105 |
| ko04932 | Non-alcoholic fatty liver disease | 9/123 | 17/737 | 0.000565 | 0.013751 | 0.012195 |
| ko03070 | Bacterial secretion system | 4/123 | 4/737 | 0.000745 | 0.01553 | 0.013772 |
| ko02024 | Quorum sensing | 5/123 | 7/737 | 0.001903 | 0.034734 | 0.030802 |
| ko00900 | Terpenoid backbone biosynthesis | 6/123 | 11/737 | 0.00432 | 0.066492 | 0.058966 |
| ko00906 | Carotenoid biosynthesis | 3/123 | 3/737 | 0.004554 | 0.066492 | 0.058966 |

those in condensin and cohesin, are conserved in these genomes, chromatin folding and chromosome condensation in these green algae occur without the linker histone H1.

The *M. hakoo* genome has a very high G + C content and abundant G4. The G4 structure, which is commonly located in telomere sequences but is also present within chromosomes, has diverse functions, including transcriptional and translational regulation. We speculate that a high frequency of the G4 consensus sequence is a characteristic of the *M. hakoo* genome, and G4-related biological processes may have contributed to the elevated genomic G + C content in this species during its evolution.

Principal component analysis of the orthogroup composition of *M. hakoo* and 14 other microalgal species resolved multiple groups of species. Interestingly, *M. hakoo* and *Ostreococcus*, which have extremely small genomes, were placed in separate groups. This result indicates that ortholog compositions in microalgae are not dependent on the genome size but rather may reflect lineage-specific gene gains/losses. The AS gene set reflects well the genomic, metabolomic, and physiological characteristics of microalgae. For example, the AS gene set included those genes associated with terpenoid-related secondary metabolites. Carotenoids, one subgroup of tetraterpenoids, play a role as an antenna pigment for harvesting light and provide protection against oxidative stress[52,53], which is beneficial for human health. Large-scale production of carotenoids for health-related industries using microalgae is flourishing[54]. The AS gene set determined in the current study provides information relevant for bioengineering of microalgae. In addition to the *M. hakoo* genome described in this study, the availability of genome sequences for a broad range of other ultrasmall algae would provide a foundation to identify the minimal conserved gene set of plants (algae and land plants), and to understand how photosynthetic eukaryotes thrive in diverse environments.

## Methods

**Materials.** *Medakamo hakoo* 311 was obtained from the personal aquarium of Prof. Kuroiwa (Kagurazaka, Tokyo, Japan)[4]. The *M. hakoo* strain was cultured in 0.05% HYPONeX (HYPONeX Japan Corp., Ltd., Osaka, Japan) liquid medium and on 0.05% HYPONeX gellan gum-based solid medium in plates. *Cyanidioschyzon merolae* 10D (Toda et al. 1995) was cultured in Misumi–Kuroiwa medium at pH 2.2 and 42 °C[55]. The Misumi–Kuroiwa medium was prepared by diluting 1 mL of a commercial nutrient solution (Hyponex, N: P: K 10: 8: 8; Hyponex Japan, Osaka, Japan) to 1 L with distilled or tap water. The pH in the medium was adjusted to pH 2.2 with 1 mL concentrated HCl. Diploid *Saccharomyces cerevisiae* BY4743 strains were cultured at 30 °C in YPD medium that contained 1% yeast extract (Oriental Yeast Co., Ltd., Tokyo, Japan), 2% peptone (Kyokuto Co., Ltd., Tokyo, Japan), and 2% glucose[56]. The *B. braunii* Kützing (NIES-2199) line was obtained from the Microbial Culture Collection at the National Institute for Environmental Studies (Japan) and cultured in AF-6 medium[57] at 22 °C.

**Synchronization culture.** The light–dark cycle for the cell-cycle synchronization culture was as follows: 12-h light:12-h dark. In the mitotic phase, *M. hakoo* cells were sampled every hour and examined using a microscope. Each cell type (one-cell, two-cells, and four-cells) was counted. More than five fields of view ($1 \times 10^3$ μm²) were selected for each sample. This experiment was performed several times and representative results are presented.

**Fluorescence microscopy.** The *M. hakoo*, *C. merolae*, and *S. cerevisiae* cells were stained with 4', 6-diamidino-2-phenylindole (DAPI) and SYBR Green I (Molecular Probes, Eugene, OR, USA)[5]. SYBR Green I stain, which has been used to examine cell nuclei in various algae because it is unaffected by the genomic G + C content[58], was used to confirm the presence of the cell nuclei and organelle nucleoids revealed by conventional DAPI staining[59]. Cultures were centrifuged and the resulting pellet was resuspended, after which a 3 μL aliquot of the solution was placed on a glass slide to form a droplet. Next, 3 μL 1% (v/v) glutaraldehyde in NS buffer (0.25 M sucrose, 1 mM EDTA, 7 mM 2-mercaptoethanol, 0.8 mM PMSF, 1 mM magnesium chloride, 0.1 mM calcium chloride, 0.1 mM zinc sulfate and 20 mM Tris-HCl, pH 7.6) was added to the droplet, followed by the addition of 3 μL DAPI (15 μg mL⁻¹) or 3 μL SYBR Green I (1 μg mL⁻¹). A coverslip was placed on the droplet and then gently pressed. The stained samples were observed using an Olympus BH-2 BHS epifluorescence microscope.

**Transmission electron microscopy.** Electron microscopy analyses were performed as previously described[4]. Briefly, *M. hakoo* was fixed for 4 h in 1% (v/v) glutaraldehyde in a sodium cacodylate buffer. After post-fixation and dehydration steps, the samples were embedded in Spurr's resin. Ultrathin sections of the samples were stained with 5% (w/v) uranyl acetate and lead citrate. The JEM 1200 EXS electron microscope (JEOL Ltd., Tokyo, Japan) was used to examine the prepared samples.

***De novo* whole-genome assembly.** *Medakamo hakoo* cells were frozen and then ground using a mortar and pestle. We extracted genomic DNA in two phenol extraction cycles, which were followed by an ethanol precipitation step. The DNA solution was purified by cesium chloride density-gradient centrifugation. The genomic DNA was purified using the AMPure XP kit (Beckman Coulter, CA, USA). Purified samples were fragmented using g-TUBE (Covaris, IL, USA). The fragmented DNA was blunted and fused with SMRTbell adapters using the SMRTbell Template Prep Kit 1.0 (Pacific Bioscience, CA, USA). We evaluated the size distribution of the adapter-fused DNA by pulse-field electrophoresis and

performed a size selection step (15.0 kb cut-off) using the BluePippin system (Sage Science, MA, USA). The sequencing library was quantified using the Agilent 2200 TapeStation (Agilent Technologies, CA, USA). The sequencing primer was annealed and DNA polymerase was bound to the sequencing library using the DNA/Polymerase Binding Kit P6 (version 2) (Pacific Bioscience, CA, USA). The sequencing templates were bound to magnetic beads using MagBead One-CellPerWell (Pacific Bioscience, CA, USA) and added to a SMART cell for the subsequent sequencing on the PacBio RS II system (Pacific Bioscience, CA, USA). For the *de novo* sequence assembly, the obtained reads were analyzed with the RS_HGAP Assemble.3 program of the SMRT analysis software (version 2.3.0).

**RNA-seq analysis.** *Medakamo hakoo* cells were ground to a powder in liquid nitrogen with a pestle and mortar. The cell powder was resuspended in 5 mL warmed (55 °C) nucleic acid extraction buffer (300 mM NaCl, 50 mM Tris-HCl [pH 7.6], 100 mM EDTA [pH 8.0], 2% Sarkosyl, and 4% SDS) and the resulting solution was stirred. The RNA extraction using a phenol/chloroform/isoamyl alcohol mixture (25:24:1, v/v/v) was repeated twice. The extract was purified by ethanol precipitation, and the total RNA was isolated using the RNeasy Plant Mini Kit (Qiagen, CA, USA). Next, 1 μL (5 units) Recombinant DNase I (RNase-free) (Takara Bio, Shiga, Japan) in 10 μL 10× buffer and 1 μL (40 units) Recombinant RNase Inhibitor (Takara Bio) were added to 100 μL crude RNA solution, which was then incubated at 37 °C for 30 min. After the addition of 5 μL 0.5 M EDTA, the solution was incubated at 80 °C for 5 min. The RNA in the solution was precipitated in ethanol and then dissolved in pure water. After the quality was checked using a NanoDrop spectrophotometer (Thermo Fisher Scientific, MA, USA) and the Agilent 2200 TapeStation, a sequencing library was produced using the TruSeq Stranded mRNA Sample Prep Kit (Illumina, CA, USA). The quality of the sequencing library was evaluated using the Agilent 2100 Bioanalyzer (Agilent Technologies). Both cBot and the HiSeq PE Cluster Kit (version 4) (Illumina) were used for the cluster formation step. The library was sequenced (100-bp paired-end reads) using the HiSeq 2500 system and the HiSeq SBS Kit (version 4) (Illumina).

**Annotation of the *M. hakoo* genome.** Sixteen contigs were annotated after removing two organellar contigs. The completeness of the contigs was evaluated using BUSCO (version 5.2.2)[19] and the chlorophyta_odb10 dataset. Repeat sequences were identified and masked using RepeatModeler (version 2.0.2)[60] and RepeatMasker (version 4.1.1) (https://www.repeatmasker.org/). Gene models were predicted according to RNA-seq reads, which were trimmed using the default options of fastp (version 0.20.0)[61]. The trimmed RNA-seq reads were mapped to the contigs using the default options of HISAT2 (version 2.2.1)[62], whereas the initial gene models were predicted using "stopCodonExcludedFromCDS = False" of BRAKER2 (version 2.1.5)[21]. Gene models were also predicted using PASA (version 2.4.1)[63], GeneMark-ET (version 4.33)[64], and SNAP (version 2006-07-28)[65] in the funannotate pipeline (version 1.8.9) (https://github.com/nextgenusfs/funannotate). These gene models were combined with those obtained from BRAKER2 (with weight = 1) using EvidenceModeler (version 1.1.1)[66]. The quality of gene prediction was evaluated using BUSCO[19,20] with the protein mode and the chlorophyta dataset. Functional annotation was conducted using eggNOG-mapper 2.1.9[22] and GhostKOALA[23].

**Assessment of the quality of the *M. hakoo* genome assembly and annotation.** To evaluate the quality of the *M. hakoo* genome assembly, we performed a BUSCO analysis (Simão et al. 2015) using the chlorophyta dataset. Contig sequences excluding organelle-derived contigs and predicted protein-coding sequences were used for the BUSCO analysis.

**Phylogenomic analyses of amino acid datasets.** The chloroplast genomes of *M. hakoo* and 63 green algae reported by Lemieux et al.[67] were included in the phylogenomic analyses. A total of 79 protein-coding genes were used to construct the following phylogenetic datasets as previously: accD, atpA, B, E, F, H, I, ccsA, cemA, chlB, I, L, N, clpP, cysA, T, ftsH, infA, minD, petA, B, D, G, L, psaA, B, C, I, J, M, psbA, B, C, D, E, F, H, I, J, K, L, M, N, T, Z, rbcL, rpl2, 5, 12, 14, 16, 19, 20, 23, 32, 36, rpoA, B, C1, C2, rps2, 3, 4, 7, 8, 9, 11, 12, 14, 18, 19, tufA, ycf1, 3, 4, 12, 20, 47, and 62. Amino acid datasets were prepared as follows. The deduced amino acid sequences of the 79 selected genes were aligned using MUSCLE 3.8[68]. The ambiguously aligned regions in each alignment were removed using TRIMAL 1.4[69] with the following settings: block = 6, gt = 0.7, st = 0.005, and sw = 3. The protein alignments were concatenated using Phyutility 2.2.6[70]. Maximum-likelihood analyses were conducted using the edge-linked partition model of IQ-TREE 1.6.1[71,72]. The datasets were partitioned by gene, with the model applied to each partition. The optimal amino acid substitution model for each gene, partitioned by the datasets, was selected according to the Bayesian information criterion using the ModelFinder function of IQ-TREE[73]. Branch support for the resulting ML trees was calculated via a non-parametric bootstrap analysis and the SH-like approximate likelihood-ratio test[74]. Bayesian analyses were performed using the site-heterogeneous CATGTR + Γ4 model and PhyloBayes 4.1[75]. Five independent chains were run for 5000 cycles, and consensus topologies were calculated from the saved trees using the BPCOMP program of PhyloBayes after a burn-in of 1250 cycles. The largest discrepancy value across all bipartitions in the consensus

topologies (maxdiff) under these conditions was less than 0.13, suggesting that the chains were substantially converged.

The sequences of the chloroplast Rubisco large subunit gene (*rbcL*) of the algal strains identified as *Choricystis* and *Botryococcus* species were obtained as aligned sequences following a BLASTN search of the NCBI database (https://www.ncbi.nlm.nih.gov/) using the *M. hakoo rbcL* sequence (1431 bp) (accession no. LC709230) as the query. Sequences were aligned using ClustalX[76]. Additionally, the *rbcL* sequences of SAG 251-1 (NIES-1436) and SAG 251-2[18] were determined by Sanger sequencing of the PCR products (accession nos. LC709231 and LC709232) and then added to the alignment. The ML analyses of the aligned *rbcL* sequences were performed using MEGAX[77], with the best-fit model (GTR + G + I) selected by MEGAX and topological support assessed with 1000 bootstrap replicates[78]. Three *Botryococcus* sequences were treated as the outgroup on the basis of the present chloroplast multigene phylogeny (Fig. 2b).

**Lipid formation culture.** We cultured *M. hakoo* in the liquid media described in Supplementary Table 8. For *B. braunii*, AF-6 was used as the normal medium. After a 13-day culture, cells were collected by centrifugation and then stained with 20 μg mL$^{-1}$ Nile Red diluted with phosphate buffer.

**Gene expression analysis.** We used the Genedata Profiler Genome software (version 10.1.14a; Genedata, Basel, Switzerland) to analyze the assembled genomic sequence and annotation data. TopHat (version 2.0.14)[79,80] was used for mapping. The total read count was 68,289,325 and 95.9% of the reads were mapped onto the *M. hakoo* genome.

**Genome size and gene number comparison among plants.** Genomic data (nuclear and organellar genomes) were obtained from the RefSeq database[81]: *A. thaliana, Glycine max, Oryza sativa, Selaginella moellendorffii, Physcomitrella patens, Amborella trichopoda, C. reinhardtii, V. carteri* f. *nagariensis, Monoraphidium neglectum, O. lucimarinus* CCE9901, *O. tauri, B. prasinos, Micromonas commoda, Micromonas pusilla* CCMP1545, *C. subellipsoidea* C-169, *C. variabilis, A. protothecoides, C. merolae* strain 10D, *G. sulphuraria*, and *C. crispus*. The *Chloropicon primus* genome was previously analyzed by Lemieux et al.[82]. The genome size and gene number analyses did not include organellar genome data.

**Pathway analysis.** A BLASTP analysis was performed by screening the KEGG database using the predicted CDSs in the *M. hakoo* genome as queries. The K numbers of each gene were obtained. Next, the pathway count data for each organism (*C. reinhardtii, V. carteri* f. *nagariensis, O. lucimarinus* CCE9901, *M. commoda, C. subellipsoidea* C-169, *C. variabilis*, and *A. protothecoides*) were acquired from the KEGG database (https://www.genome.jp/kegg/kegg_ja.html) and compared with the pathway information for the *M. hakoo* genome.

**Enrichment analysis.** A KEGG pathway enrichment analysis of the common gene sets was performed using enrichKEGG in the clusterProfiler package[83]. The K numbers of the C15 and PS gene sets were used for this analysis. The C15 gene set served as the background to assess the KEGG pathway enrichment of the AS gene set. Additionally, "ko" was selected as the parameter for "organisms". Details of the statistics for the enrichment analysis are described in the Statistics and Reproducibility section.

**SDS-PAGE and in-gel digestion.** Protein samples were dissolved in the sample buffer and partially separated (approximately 1 cm) using a NuPAGE Bis-Tris gel (Thermo Fisher Scientific, CA, USA). Electrophoresis was performed according to instructions of the manufacturer. Each lane was excised from the unstained gel. In-gel digestion was performed using 0.01 μg/μL LysC and trypsin[84].

**Mass spectroscopic and chromatographic methods, instrumentation, and database searches.** The resulting peptides were analyzed using the Q Exactive hybrid mass spectrometer (Thermo Fisher Scientific, CA, USA)[85]. The MS/MS spectra were interpreted and then peak lists were generated using Proteome Discoverer 2.2.0.388 (Thermo Fisher Scientific, CA, USA). The SEQUEST program was used to search the in-house *M. hakoo* protein database with the following settings: enzyme selected with up to two missing cleavage sites; peptide mass tolerance, 10 ppm; MS/MS tolerance, 0.02 Da; fixed modification, carbamidomethylation (C); and variable modification, oxidation (M). Peptides were identified according to significant Xcorr values (high confidence filter). The peptide identification and modification information obtained from SEQUEST was manually examined and filtered to obtain confirmed peptide identification and modification lists for the HCD MS/MS analysis. The precursor ion intensity (normalized against the total peptide amount) was used for the label-free quantification.

**Codon count.** The detected peptides encoded by specific regions in the *M. hakoo* genome were used for the codon count. All complete CDSs mapped on the basis of these peptides were used. We counted the codons in the genome sequence

encoding the peptides using the R package coRdon (https://github.com/BioinfoHR/coRdon).

**G4 analysis**. The complete *M. hakoo* draft genome sequence and the *Escherichia coli*, *C. reinhardtii*, *C. merolae*, and *S. coelicolor* genome sequences were analyzed using the default settings of the pqsfinder software to identify potential G4-forming sequences[27].

**Orthogroup analysis**. The *C. merolae*[6,7], *O. tauri*[9,10], *S. cerevisiae* strain S288C[86], and *M. hakoo* amino acid sequence datasets were analyzed. For the orthologous group analyses, gene families were identified using OrthoFinder[40]. Protein sequences were compared in all-*vs*-all BLASTP searches using the NCBI blast+ toolkit[87], as suggested in the OrthoFinder manual.

**Principal component analysis**. Orthogroup composition data (Supplementary Data 4) for microalgae were analyzed with the prcomp package (version 4.0.2) in R.

**Nomenclatural Acts**. This published work and the nomenclatural acts (Supplementary Note 1) it contains have been registered in PhycoBank, the proposed online registration system for the International Code of Nomenclature for algae, fungi and plants (ICN). The PhycoBank LSIDs (Life Science Identifiers) can be resolved and the associated information viewed through any standard web browser by appending the LSID to the prefix "http://phycobank.org/". The LSIDs for this publication are: 103506; 103507; 103508.

**Statistics and reproducibility**. All of the culture experiments presented in this paper have been conducted multiple times to confirm reproducibility. To analyze the table data and draw the figures, we used the tidyverse package (version 1.3.1) in R and pandas (version 1.0.5) in python. Brunner-Munzel test was performed with lawstat package (version 3.5) in R. The stats package (version 4.0.2) in R was used for Bonferroni correction of *p*-values. In the gene enrichment analysis, the *p*-values were calculated using a hypergeometric distribution, and the *p*-values of each pathway were adjusted according to the Benjamini–Hochberg method[88].

**Reporting summary**. Further information on research design is available in the Nature Portfolio Reporting Summary linked to this article.

## Data availability

The source data underlying Figs. 1h, 3b, e, 4b, c, 5b–d are provided as Supplementary Data 6. The genome sequence read data were deposited in the Sequence Read Archive (accession numbers: SRR16480670–SRR16480673). The assembled chromosomal DNA sequences were deposited in GenBank (accession numbers: CP089450–CP089465). The transcriptome sequencing data were deposited in the Sequence Read Archive (accession number: SRR19165385), whereas the proteome data were deposited in the jPOST repository (accession number: JPST001585). All other data are available from the corresponding author.

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

## Acknowledgements

This research was supported by MXT/JSPS KAKENHI funding to T.K. (19H03260 and 22H02657) and S. Matsunaga (20H05911). It was also supported by JST-CREST (JPMJCR20S6) and JST-OPERA (JPMJOP1832) grants to S. Matsunaga. We thank Edanz (https://jp.edanz.com/ac) for editing a draft of this manuscript.

## Author contributions

S.K., O.M., S. Matsunaga, and T.K. designed the project. H.K. and T.K. performed the morphological analyses of *M. hakoo*, *C. merolae*, and *S. cerevisiae*. S.K., S.S., H.Y., and M.K. analyzed the nuclear genome data. M.T. analyzed the organellar genome data. H.N. and S. Maruyama performed the phylogenetic analysis. O.M., Y.T.I., and M.T. extracted the genomic DNA. Y.T.I., K.K., S.N., and Y.M. performed the proteome analysis. T.K. and H.K. performed the optical and electron microscopy analyses. Y.T.I. and T.M.M. performed the lipid accumulation assay. S.K., S. Maruyama, T.S., N.I., Y.O., F.Y., and S. Matsunaga analyzed the conserved genes in *M. hakoo*. S.K., S. Matsunaga, S. Maruyama, H.N., M.T., T.S., N.I., and Y.O. wrote the manuscript, which was edited by the other authors.

## Competing interests

The authors declare no competing interests.
