## [Peer Review File · Communications Biology]

Reviewers' comments:

Reviewer #1 (Remarks to the Author):

The authors describe extensive genomic analyses of the ultrasmall green alga *Medakamo hakoo*. Their work will be a valuable asset for the study of algal biology. However, I have significant reservations about recommending the work in its current form. These mainly stem from the authors' claim that the species has the fewest genes of any Viridiplantae species. The reason for doubt here lies in the methods they used to come to this conclusion. They estimated gene count from RNAseq mapping, Tophat, and Augustus gene prediction using *Chlamydomonas* as a reference organism. In my own experience, Augustus will severely underestimate the number of genes in closely related green algae, and *M. hakoo* is not so closely related to *Chlamydomonas*. In my own experience with using Augustus for Trebouxiophyceae species, a large number of genes were not found or not modeled correctly with August or Maker(2).

The high-coverage PacBio assembly should provide the basis for accurate gene prediction; however, I do not believe the methods used for gene prediction and modeling are sufficient to support their claims. Still, their claim that *M. hakoo* has the smallest genome among freshwater Viridiplantae species (discovered to date) may still hold water.

In summary, it is highly likely that the gene modeling algorithms used by the group are underestimating the true gene counts in *M. hakoo*'s genome. I recommend that they use other software to predict genes with their RNAseq evidence and also use ab initio approaches (since many genes may not be expressed in the conditions tested). For example, BRAKER2, GNOMON, or even genes predicted using the HISAT2-STRINGTIE-BALLGOWN pipeline should be compared for cross validation. As I have stated, Augustus will underestimate total gene counts for even closely related green algae, let alone a Trebouxiophyte. The authors should perform more robust analyses or pare back their claims. If these can be addressed, the paper should make a valuable contribution to the field and Comms Bio.

Some minor comments:

Line 536: The authors state that the genome size did not include organellar genome data, but in line 112 they state that some contigs could be organellar sequences. I would suggest to include all of the DNA sequence of the organism as its 'genome' and make distinctions between nuclear, chloroplast, and mitochondrial genomes whenever possible (especially in the abstract).

Reviewer #2 (Remarks to the Author):

In this manuscript, the authors describe the genome of the ultrasmall green alga *Medaka hakoo*. Overall, the manuscript is interesting, but I have concerns about the rigor of some of the analyses and about the corresponding conclusions that I would like to see addressed before this manuscript is considered for publication. Therefore, my recommendation is major revision.

In the manuscript, the authors make many observations/conclusions based on the genes that are present/absent in *M. hakoo*. However, I am concerned that the annotation of this genome might be missing several genes due to the methods employed:

1. In the methods, the authors state that they used RNAseq data to help annotate the genome. They further mentioned on page 18 that the cell cultures were synchronized. Was the RNAseq data extracted from these synchronized cultures? Was total RNA isolated from a single time point or was it

isolated from multiple time points? If the total RNA was isolated from synchronized cultures at a single time point, genes that are not expressed at that time point under the culture conditions will not be present in the RNAseq dataset.

2. On page 21, the authors used a quality score cutoff of 20 ($QV > 20$) to filter the RNAseq data. This cutoff is far from stringent. I expected a cutoff of at least 28. Cutadapt tends to leave adapters in the data and the FASTX-Toolkit has not been updated in over a decade. I recommend using FASTP instead.

3. The authors have access to RNAseq data, yet they chose the *Chlamydomonas* model to annotate the *M. hakoo* genome. The *Chlamydomonas reinhardtii* genome is about 120 Mbp, or about 10 times the size of *M. hakoo*. It is very unlikely that the *Chlamydomonas* model will serve as a good model for *M. hakoo*. Why not use BRAKER to annotate the *M. hakoo* genome directly from its RNAseq data? BRAKER will create a custom Augustus model directly from the *M. hakoo* RNAseq dataset, which should be much more suitable to its annotation. Even if the authors want to use MAKER2 instead of BRAKER, they should create a *M. hakoo* model and use it for its annotation. I would not be surprised either if models from picoprasinophytes perform better than the *Chlamydomonas* one on *M. hakoo*.

4. There is no indication that the authors validated the outcome of their *M. hakoo* annotation. What if the annotations are subpar? What if they are missing several genes? The latter is very likely given the differences between the *Chlamydomonas* and *M. hakoo* genomes but even with a custom model derived from RNAseq data, a few genes might not be annotated properly (even good models can miss genes). The authors mention that they performed a BUSCO analysis in their results, but this analysis is not described anywhere in the methods. Inspecting the outcome of annotations based on the sum of multiple evidence with tools like Apollo would be valuable here. Apollo can be used to load predictions with different models and/or pipelines, RNA seq data, and the results of TBLASTN searches (see below) from reference proteomes against the genome being annotated.

5. In general, the authors make strong claims in their papers about the genes that are missing from *M. hakoo*. However, it seems that only BLASTP analyses were used to assess the presence/absence of protein coding genes. BLASTN and/or TBLASTN searches against the *M. hakoo* genome would be important: 1) genes might not have been annotated by the approach they used; 2) genes might be present but intron/exon junctions could be mis-predicted causing BLASTP searches to fail. Proving the absence of something is difficult, but the authors do not appear to have done due diligence here.

6. I could not double-check some of the claims made by the authors due to the unavailable data. The authors claim on p. 28-29 that the data will be made available in public repositories once the manuscript is published but I cannot in good conscience accept this. The accession number for the genome should already be part of this manuscript. NCBI allows for delayed data releases until a manuscript is published so there is no reason to wait. Their submission system has been overloaded with genomes backlogged for several months in the past year and waiting at the last minute is not good, especially since the process might involve a lot of debugging of the submitted annotations. I cannot and will not accept a genome manuscript without the proper accession numbers.

I'm also a bit puzzled by how the authors address the organelle genomes in the manuscript. On pages 6 and 7 the authors state: "The ends of the remaining two contigs overlapped, suggesting that these contigs form a circular chromosome (i.e., organellar genomes) (Supplementary Figs 5-7)." and "The two contigs representing the organellar genome were described previously (Supplementary Table 3, Supplementary Figs 4, 5)". Were the organelle genomes described previously or not? Overall, the corresponding paragraphs are convoluted for no reason. Of the 18 contigs, 2 are organelle genomes, the remainder form the chromosomes of the nuclear genome and are bounded by telomeric units.

On page 6, the authors state: "If the linear contigs and two contigs forming a circular chromosome were considered as the complete genome, the sequence coverage was 246.8 \times , which is sufficient for

obtaining reliable sequences, even from long-read sequencing data." This statement is poorly written and conflating the organelle genomes with the nuclear one is not logical. The genomes are not necessarily present in a 1:1:1 distribution and even if that was the case, the organelle genomes are haploid, but the nuclear genome might be haploid or diploid. It would be much better to look at the sequencing depth per contig. This would also help check for the potential presence of aneuploidy in the nuclear genome. Also, the bit "which is sufficient for obtaining reliable sequences, even from long-read sequencing data" is argumentative and should not be in the results.

On page 14, the authors compare the frequency of predicted G4 regions in *M. hakoo* to a red alga, from which they speculate that: "Thus, we speculated that a high frequency of the G4 consensus sequence is a characteristic of the *M. hakoo* genome, and G4-related biological processes may have contributed to the increase in the genomic G+C content in this species during evolution." The whole section makes absolutely no sense. Even AT rich genomes feature G-quadruplex telomeric repeats. Just because the *M. hakoo* genome assembly has more telomeric repeats than the *C. merolae* one does not mean that anything biologically significant is afoot. Changing the length of the sequencing reads and/or the software and parameters used would change the number of repeats in the assemblies. This could easily be due to assembly artefacts but even if not, differences in telomerase activity are much more likely to be important to the number of telomeric repeats than the GC contents of these genomes. The proposed speculation is not only unsubstantiated but also does not even consider the usual suspects.

On Page 15, the authors mention that genome minimizations in *M. hakoo*, *C. merolae*, and *O. tauri* were the result of separate evolutionary processes. Yet, the authors do not address what was the state of the genome of their ancestor. It appears to me that the ancestral genome was likely small. Did the authors even consider that maybe the other genomes have expanded instead?

Minor comments:

p.37 Table 1: Organelle not organella

Point-by-point responses to reviewers' comments

Reviewer #1

Comment 1: The authors describe extensive genomic analyses of the ultrasmall green alga *Medakamo hakoo*. Their work will be a valuable asset for the study of algal biology. However, I have significant reservations about recommending the work in its current form. These mainly stem from the authors' claim that the species has the fewest genes of any *Viridiplantae* species. The reason for doubt here lies in the methods they used to come to this conclusion. They estimated gene count from RNAseq mapping, Tophat, and Augustus gene prediction using *Chlamydomonas* as a reference organism. In my own experience, Augustus will severely underestimate the number of genes in closely related green algae, and *M. hakoo* is not so closely related to *Chlamydomonas*. In my own experience with using Augustus for *Trebouxiophyceae* species, a large number of genes were not found or not modeled correctly with August or Maker(2). The high-coverage PacBio assembly should provide the basis for accurate gene prediction; however, I do not believe the methods used for gene prediction and modeling are sufficient to support their claims. Still, their claim that *M. hakoo* has the smallest genome among freshwater *Viridiplantae* species (discovered to date) may still hold water. In summary, it is highly likely that the gene modeling algorithms used by the group are underestimating the true gene counts in *M. hakoo*'s genome. I recommend that they use other software to predict genes with their RNAseq evidence and also use ab initio approaches (since many genes may not be expressed in the conditions tested). For example, BRAKER2, GNOMON, or even genes predicted using the HISAT2-STRINGTIE-BALLGOWN pipeline should be compared for cross validation. As I have stated, Augustus will underestimate total gene counts for even closely related green algae, let alone a *Trebouxiophyte*. The authors should perform more robust analyses or pare back their claims. If these can be addressed, the paper should make a valuable contribution to the field and Comms Bio.

Response: Thank you very much for your valuable comments regarding our gene prediction method. We improved our gene predictions using BRAKER2, which increased the number of predicted genes from 6,399 to 7,629. Even with this increase, we are sure that the *M. hakoo* genome is smaller (in terms of the number of genes) than the genomes of green plants. However, compared with other microalgal species, the number of genes in the *M. hakoo* genome is not substantially lower. Thus, we have revised the description in the manuscript accordingly (Line 73-77).

Comment 2: Line 536: The authors state that the genome size did not include organellar

genome data, but in line 112 they state that some contigs could be organellar sequences. I would suggest to include all of the DNA sequence of the organism as its 'genome' and make distinctions between nuclear, chloroplast, and mitochondrial genomes whenever possible (especially in the abstract).

Response: Thank you for your comments. To more precisely describe the genome, we revised the manuscript so that 'genome' is used to refer to the nuclear, chloroplast, and mitochondrial genomes, whereas individual names are used to refer to each genome type.

Reviewer #2

Comment 1: In the methods, the authors state that they used RNAseq data to help annotate the genome. They further mentioned on page 18 that the cell cultures were synchronized. Was the RNAseq data extracted from these synchronized cultures? Was total RNA isolated from a single time point or was it isolated from multiple time points? If the total RNA was isolated from synchronized cultures at a single time point, genes that are not expressed at that time point under the culture conditions will not be present in the RNAseq dataset.

Response: The RNA used for the RNA-seq analysis was extracted from samples cultured under non-synchronous conditions. Therefore, the RNA used in the analysis is considered to be unbiased.

Comment 2: On page 21, the authors used a quality score cutoff of 20 ($QV > 20$) to filter the RNAseq data. This cutoff is far from stringent. I expected a cutoff of at least 28. Cutadapt tends to leave adapters in the data and the FASTX-Toolkit has not been updated in over a decade. I recommend using FASTP instead.

Response: Thank you for your important remarks regarding the sequence filtering. We have repeated the filtering using the default parameters of FASTP. As shown in the following figures, most of the reads had an average QV greater than 28. Thus, we repeated the gene prediction and transcriptome analyses using these reads.

Mean quality score distribution of filtered R1 sequence data

Mean quality score distribution of filtered R2 sequence data

Comment 3: The authors have access to RNAseq data, yet they chose the *Chlamydomonas* model to annotate the *M. hakoo* genome. The *Chlamydomonas reinhardtii* genome is about 120 Mbp, or about 10 times the size of *M. hakoo*. It is very unlikely that the *Chlamydomonas* model will serve as a good model for *M. hakoo*. Why not use BRAKER to annotate the *M. hakoo* genome directly from its RNAseq data? BRAKER will create a custom Augustus model directly from the *M. hakoo* RNAseq dataset, which should be much more suitable to its annotation. Even if the authors want to use MAKER2 instead of BRAKER, they should create a *M. hakoo* model and use it for its annotation. I would not be surprised either if models from picoprasinophytes perform better than the *Chlamydomonas* one on *M. hakoo*.

Response: Thank you for your important remarks. As mentioned above, we have reanalyzed

the data using BRAKER2 to improve the gene predictions. The manuscript text was revised accordingly.

Comment 4: There is no indication that the authors validated the outcome of their *M. hakoo* annotation. What if the annotations are subpar? What if they are missing several genes? The latter is very likely given the differences between the *Chlamydomonas* and *M. hakoo* genomes but even with a custom model derived from RNAseq data, a few genes might not be annotated properly (even good models can miss genes). The authors mention that they performed a BUSCO analysis in their results, but this analysis is not described anywhere in the methods. Inspecting the outcome of annotations based on the sum of multiple evidence with tools like Apollo would be valuable here. Apollo can be used to load predictions with different models and/or pipelines, RNA seq data, and the results of TBLASTN searches (see below) from reference proteomes against the genome being annotated.

Response: A BUSCO analysis was performed on the basis of the BRAKER2 gene prediction results. The procedure has been added to the Methods section.

Comment 5: In general, the authors make strong claims in their papers about the genes that are missing from *M. hakoo*. However, it seems that only BLASTP analyses were used to assess the presence/absence of protein coding genes. BLASTN and/or TBLASTN searches against the *M. hakoo* genome would be important: 1) genes might not have been annotated by the approach they used; 2) genes might be present but intron/exon junctions could be mis-predicted causing BLASTP searches to fail. Proving the absence of something is difficult, but the authors do not appear to have done due diligence here.

Response: As suggested, a TBLASTN analysis of the *M. hakoo* genome was performed using the RefSeq genome data of the following species as queries.

Species

Homo sapiens

Saccharomyces cerevisiae

Arabidopsis thaliana

Chlamydomonas reinhardtii

Coccomyxa subellipsoidea

Cyanidioschyzon merolae

Ostreococcus tauri

A total of 17,718 regions in the *M. hakoo* genome were detected, of which 17,106 overlapped with the gene regions identified by BRAKER2. Although the predicted genes revealed in this study are considered to be correct, we softened our claim regarding missing genes after

considering the possibility that some genes were unidentified.

Comment 6: I could not double-check some of the claims made by the authors due to the unavailable data. The authors claim on p. 28-29 that the data will be made available in public repositories once the manuscript is published but I cannot in good conscience accept this. The accession number for the genome should already be part of this manuscript. NCBI allows for delayed data releases until a manuscript is published so there is no reason to wait. Their submission system has been overloaded with genomes backlogged for several months in the past year and waiting at the last minute is not good, especially since the process might involve a lot of debugging of the submitted annotations. I cannot and will not accept a genome manuscript without the proper accession numbers.

Response: We apologize for the delay in sharing the data. We have uploaded the following data to public databases. Please check the following links. Data will be shared immediately if requested by the editor or reviewers.

Data	Repository	Accession no.	Link
Genome read	SRA	SRR16480670– SRR16480673	https://www.ncbi.nlm.nih.gov/sra/PRJNA771468
Assembled chromosomal genome	GenBank	CP089450-CP089465	To be released
RNA-seq read	SRA	SRR19165385	To be released
Proteome	jPOST	JPST001585	To be released

Comment 7: I also a bit puzzled by how the authors address the organelle genomes in the manuscript. On pages 6 and 7 the authors state: The ends of the remaining two contigs overlapped, suggesting that these contigs form a circular chromosome (i.e., organellar genomes) (Supplementary Figs 5). The two contigs representing the organellar genome were described previously (Supplementary Table 3, Supplementary Figs 4, 5). Were the organelle genomes described previously or not? Overall, the corresponding paragraphs are convoluted for no reason. Of the 18 contigs, 2 are organelle genomes, the remainder form the chromosomes of the nuclear genome and are bounded by telomeric units.

Response: A summary of the analysis of the *M. hakoo* organellar genome has already been published (Biorxiv). To avoid confusion, we have changed the description as follows:

“After the long-read sequencing analysis, we obtained 18 contigs via a de novo sequence assembly (Table 1, Supplementary Table 1). Of the 18 contigs, two were annotated as organellar genomes because they were circular sequences (Supplementary Figs 5–7). We

have published the organellar genome data in a short report available in BioRxiv²⁴. The remaining 16 contigs were chromosomal sequences flanked by telomere sequences (5'-TTAGGG-3')."

Comment 8: On page 6, the authors state: If the linear contigs and two contigs forming a circular chromosome were considered as the complete genome, the sequence coverage is sufficient for obtaining reliable sequences, even from long-read sequencing data. This statement is poorly written and conflating the organelle genomes with the nuclear one is not logical. The genomes are not necessarily present in a 1:1:1 distribution and even if that was the case, the organelle genomes are haploid, but the nuclear genome might be haploid or diploid. It would be much better to look at the sequencing depth per contig. This would also help check for the potential presence of aneuploidy in the nuclear genome. Also, the bit is sufficient for obtaining reliable sequences, even from long-read sequencing data is argumentative and should not be in the results.

Response: We removed the text, "...which is sufficient for obtaining reliable sequences, even from long-read sequencing data." We also revised the paragraph as follows:

"Next, we validated the genome assembly by confirming several benchmarks. The sequence coverage on the assembly was 246.8×. To further evaluate the assembled genome, we conducted a Benchmarking Universal Single-Copy Orthologs (BUSCO) analysis^{23,24}. After removing organellar contigs, the assembly showed 89.5% of complete BUSCOs (Fig. 2a). Also, we mapped RNA-seq reads to the contigs and most of the reads (98.6%) were aligned. Moreover, tRNAs corresponding to all 20 amino acids were identified (Supplementary Table 2)."

Comment 9: On page 14, the authors compare the frequency of predicted G4 regions in *M. hakoo* to a red alga, from which they speculate that. Thus, we speculated that a high frequency of the G4 consensus sequence is a characteristic of the *M. hakoo* genome, and G4-related biological processes may have contributed to the increase in the genomic G+C content in this species during evolution. The whole section makes absolutely no sense. Even AT rich genomes feature G-quadruplex telomeric repeats. Just because the *M. hakoo* genome assembly has more telomeric repeats than the *C. merolae* one does not mean that anything biologically significant is afoot. Changing the length of the sequencing reads and/or the software and parameters used would change the number of repeats in the assemblies. This could easily be due to assembly artefacts but even if not, differences in telomerase

activity are much more likely to be important to the number of telomeric repeats than the GC contents of these genomes. The proposed speculation is not only unsubstantiated but also does not even consider the usual suspects.

Response: A high G+C content was detected at the telomeres as well as inside contigs (e.g., CDS), suggesting that G-quadruplex sequences accumulated throughout the genome. We believe that the accumulation of G-quadruplex consensus sequences in the *M. hakoo* genome may be biologically significant because it has not been observed in other organisms. We have revised the text in the relevant section to avoid confusion.

Comment 10: On Page 15, the authors mention that genome minimizations in *M. hakoo*, *C. merolae*, and *O. tauri* were the result of separate evolutionary processes. Yet, the authors do not address what was the state of the genome of their ancestor. It appears to me that the ancestral genome was likely small. Did the authors even consider that maybe the other genomes have expanded instead?

Response: As you point out, it is possible that speciation occurred and these organisms maintained their small genomes. We have corrected the text as follows to indicate this possibility:

“Although the mechanisms underlying the maintenance of the minimal genome and the small cells of *C. merolae*, *O. tauri*, and *M. hakoo* are unknown, generic factors, such as photosynthetic efficiency, may have driven the genome to be smaller or to suppress the expansion of the genome size.”

Comment 11: p.37 Table 1: Organelle not organella

Response: The table was corrected.

Reviewers' comments:

Reviewer #1 (Remarks to the Author):

I am generally satisfied with the authors revisions. Still, I think the data should have been submitted instead of being only available upon request. Pending the validation of source data, I could recommend the work for publication.

Reviewer #2 (Remarks to the Author):

The authors produced a revised version of their manuscript on the genome of the green alga *Medakamo hakoo* in which the methods have been greatly improved. Unfortunately, however, the manuscript needs a thorough rewrite/streamlining for clarity and I cannot recommend publication in its current form as it suffers both from structural and content issues. Also, 6932 out of the 7628 (90.1%) of the *M. hakoo* proteins are annotated as hypothetical proteins in the provided file (10805_1_supp_369744_rccbnr.csv) which does raise questions about the overall quality of the functional genomic inferences performed therein (the authors will want to check that carefully). While I can appreciate the amount of work that went in this study, my recommendation is to reject this manuscript so that the authors can take their time to rewrite it properly and resubmit it at a later stage.

Structural issues

In their results section, the authors describe the alga, describe (somewhat) its nuclear genome but then talk about phylogenetic inferences derived from the plastid genome in the same section!?, then they talk about the genome some more, then they talk again about the alga description....

The authors should: 1) formally describe this alga: 2) position it within the green algae: 3) then, and only then, talk about its nuclear genome. This would make for a much clearer story, e.g. Taxonomic description, phylogenetic classification, and genome sequencing of the ultrasmall green alga *Medakamo hakoo*. More importantly, the authors should first describe the *M. hakoo* nuclear genome properly before doing any comparison. As written, the description of the *M. hakoo* nuclear genome structure and/or physical properties is scattered all over the place. For example, one of its most salient features, a 73% GC content for a freshwater organism, is only introduced on page 12 and I had to jump back and forth between pages of this manuscript to read about the genome. This should not happen in a manuscript with such a focus on genomics.

Also, the authors already published the organelle genomes of *M. hakoo* as a separate manuscript in BioRxiv so there is no reason to describe or include them here (supplementary figs 5 & 6, supplementary table 3). The BioRxiv was clearly meant as a separate manuscript so the information should not be duplicated. On lines 183 -185, the authors state: "Our data revealed the conserved photosynthesis-related genes in *M. hakoo*, including genes encoding subunits of photosystems I and II, the cytochrome b6/f complex, plastid ATP synthase, and the Calvin cycle enzymes." Most if not all of these are encoded in the chloroplast genome and have been known for decades. This has no place in a manuscript describing the nuclear genome. If the authors really want to include the organelle genomes here, there should be clear distinctions between the mitochondrial, chloroplast and nuclear genomes.

Content issues

The main issue with this manuscript is the heavy focus on the comparison between *M. hakoo* and *Arabidopsis* and their emphasis on plant metabolism while trying to frame that under a "minimum genome" narrative. The introduction focuses heavily on Archaeplastida and the authors do compare to the red algae *C. merolae* so I'm assuming that when they say plants what they really mean is all plants including algae (or at least sometimes), but most readers will automatically think land plants

(especially when *Arabidopsis* is thrown in the mix) and the manuscript is often very confusing. Yes, algae are part of Viridiplantae but they are not land plants and comparing a trebouxiophyte (which is not the most basal lineage of green algae) to a dicot that is not even close to basal among land plants while ignoring nearly everything about prasinophytes and charophytes just doesn't make a lot of sense.

For examples, on lines 66-68 the authors state: "Microalgae usually have a simple genome and cell structure, but they have retained the key features of plants, including photosynthetic activities and plant hormone production". Problem is, algae did not evolve from land plants so that statement is just plain wrong (land plants evolved from algae, not the other way around) unless the authors mean all plants including algae, but then the last common ancestor of algae was 1) photosynthetic and 2) likely very small so what are they trying to say here??? On lines 326 – 329: "Although these algae have relatively few genes and a small genome, they inhabit very different environments and are not phylogenetically related. Thus, their genome minimizations were probably the result of separate evolutionary processes." The authors again totally ignore the elephant in the room in that the ancestral algal genome was very likely small; did the authors even check the distribution of genomes sizes in a phylogenetic framework?

Lines 68-69: "Thus, the minimal microalgal genome may provide clues regarding the core gene set required by plants". Plants or land plants? *M. hakoo* is on a completely different branch of the Viridiplantae than *Arabidopsis* and looking at charophytes would be a lot more informative for that purpose. Sure *M. hakoo* may not require as many housekeeping genes but that knowledge will not translate to land plants. If the goal is to find the minimal set of genes for all plants (not land plants), then focusing on protists is the way to go. Throwing in *Arabidopsis* just leads to more confusion (+ *Physcomitrella patens* appears out of the blue in Figure 3 !?). More importantly, a core set of genes is defined as the minimum set required for an organism to function, so the authors are basically saying here that finding the core set of genes will help find the core set of genes!

This dubious and/or confusing logic is pervasive throughout the manuscript. For example, on lines 226 -228: "Moreover, similar to *C. merolae*, *M. hakoo* lacks plant-specific NSE5 and NSE6 orthologs (Supplementary Table 5), which are the non canonical and evolutionarily non-conserved components of the SMC5/6 complex. In *A. thaliana*...". Plant specific as in land plants? If so, why would anyone expect them in algae? If they are non-canonical, why expect them at all? Or course more complex organisms will have evolved more molecular complexity over time. On lines 232-232: "Accordingly, it is possible that SMC5/6 is more functionally limited in *M. hakoo* than in flowering plants." Again, why compare to land plants???

Overall, the manuscript is very verbose yet contains very little information at times. For example, on lines 217 – 220 "Our analysis confirmed that canonical components of all SMC complexes are conserved in *M. hakoo* (Supplementary Table 4), suggesting that SMC complexes are indispensable for the chromosome maintenance in *M. hakoo* and in other plant species." A basic set of SMC proteins is essential in eukaryotes. No chromosome maintenance? Death. This sentence is not informative and setups the meaningless SMC5/6 comparison with *Arabidopsis*. Again, on lines 250 - 253 "Therefore, our results suggest that mid-SUN is a fundamental INM protein and the acquisition of diverse factors enabled the development of relatively sophisticated and complex chromatin structures in phanerophytes." More complex organisms, more complex machinery? How is that informative? Why is that even in the RESULTS section. This is not a result.

Other examples of confusion: table 2 Comparison of the genome data for microalgae => *Arabidopsis* is not an alga... Lines 178 – 180 "In summary, we revealed that *M. hakoo* has one of the smallest genomes among Viridiplantae species, and its gene sets that regulate the central dogma..." What dogma? This makes no sense. Lines 315-318 "Thus, we speculated that a high frequency of the G4 consensus sequence is a characteristic of the *M. hakoo* genome, and G4-related biological processes may have contributed to the increase in the genomic G+C content in this species during evolution."

Why would telomere repeat units influence the full GC content of an organism? There is no basis for this speculation, this is not a result and certainly should not be written in the RESULTS section, and there is no explanation/hypothesis whatsoever about any mechanism that would explain this speculation.

Reviewer #2 (Remarks to the Author):

Comment 1: The authors produced a revised version of their manuscript on the genome of the green alga *Medakamo hakoo* in which the methods have been greatly improved. Unfortunately, however, the manuscript needs a thorough rewrite/streamlining for clarity and I cannot recommend publication in its current form as it suffers both from structural and content issues.

Response: Thank you very much for your positive assessment of our revisions. We have gone through the entire manuscript again and rewritten it according to your suggestions.

Comment 2: Also, 6932 out of the 7628 (90.1%) of the *M. hakoo* proteins are annotated as hypothetical proteins in the provided file (10805_1_supp_369744_rccbnr.csv) which does raise questions about the overall quality of the functional genomic inferences performed therein (the authors will want to check that carefully). While I can appreciate the amount of work that went in this study, my recommendation is to reject this manuscript so that the authors can take their time to rewrite it properly and resubmit it at a later stage.

Response: There appears to be some misunderstanding about what is presented in 10805_1_supp_369744_rccbnr.csv. The provided information indicates approximately 80.7% of the CDSs have been annotated on the basis of InterProScan and eggNOG.

Comment 3: In their results section, the authors describe the alga, describe (somewhat) its nuclear genome but then talk about phylogenetic inferences derived from the plastid genome in the same section, then they talk about the genome some more, then they talk again about the alga description....

Response: To address this comment, we have rearranged the Results section. Specific details regarding our changes are provided below.

Comment 4: The authors should: 1) formally describe this alga: 2) position it within the green algae: 3) then, and only then, talk about its nuclear genome. This would make for a much clearer story, e.g. Taxonomic description, phylogenetic classification, and genome sequencing of the ultrasmall green alga *Medakamo hakoo*. More importantly, the authors should first describe the *M. hakoo* nuclear genome properly before doing any comparison. As written, the description of the *M. hakoo* nuclear genome structure and/or physical properties is scattered all over the place. For example, one of its most salient features, a 73% GC content for a freshwater organism, is only introduced on page 12 and I had to jump back and forth between pages of this manuscript to read about the genome. This should not happen in a manuscript with such a focus on genomics.

Response: As suggested, in the revised manuscript, we first describe the phylogenetic classification of *Medakamo hakoo*. We then provide a taxonomic description and information regarding the genome sequence. Additionally, we describe the nuclear

genome before mentioning the comparison.

Comment 5: Also, the authors already published the organelle genomes of *M. hakoo* as a separate manuscript in BioRxiv so there is no reason to describe or include them here (supplementary figs 5 & 6, supplementary table 3). The BioRxiv was clearly meant as a separate manuscript so the information should not be duplicated.

Response: As suggested, we removed Supplementary Figures 5, 6 and 7 as well as Supplementary Table 3.

Comment 6: On lines 183-185, the authors state: "Our data revealed the conserved photosynthesis-related genes in *M. hakoo*, including genes encoding subunits of photosystems I and II, the cytochrome b6/f complex, plastid ATP synthase, and the Calvin cycle enzymes." Most if not all of these are encoded in the chloroplast genome and have been known for decades. This has no place in a manuscript describing the nuclear genome. If the authors really want to include the organelle genomes here, there should be clear distinctions between the mitochondrial, chloroplast and nuclear genomes.

Response: We have revised the statement so it is now as follows:

"Although almost all fundamental photosynthesis-related genes were conserved in *M. hakoo*, genes encoding stress-related light-harvesting complex (LHC)-like proteins were not detected in the *M. hakoo* genome (Supplementary Figures 7, 8)." (Line 227 - 229)

Comment 7: The main issue with this manuscript is the heavy focus on the comparison between *M. hakoo* and *Arabidopsis* and their emphasis on plant metabolism while trying to frame that under a "minimum genome" narrative. The introduction focuses heavily on Archaeplastida and the authors do compare to the red algae *C. merolae* so I'm assuming that when they say plants what they really mean is all plants including algae (or at least sometimes), but most readers will automatically think land plants (especially when *Arabidopsis* is thrown in the mix) and the manuscript is often very confusing. Yes, algae are part of Viridiplantae but they are not land plants and comparing a trebouxiophyte (which is not the most basal lineage of green algae) to a dicot that is not even close to basal among land plants while ignoring nearly everything about prasinophytes and charophytes just doesn't make a lot of sense.

Response: To address your comment, we have removed the sentences describing the comparative analysis with *Arabidopsis thaliana* (including NSE5/6 and SMC5/6 proteins) from the manuscript. Additionally, Supplementary Table 4 in the previously submitted manuscript has been deleted.

Comment 8: For examples, on lines 66-68 the authors state: "Microalgae usually have a simple genome and cell structure, but they have retained the key features of plants, including photosynthetic activities and plant hormone production". Problem is, algae did not evolve from land plants so that statement is just plain wrong (land plants evolved from

algae, not the other way around) unless the authors mean all plants including algae, but then the last common ancestor of algae was 1) photosynthetic and 2) likely very small so what are they trying to say here.

Response: We are aware that land plants evolved from algae and not *vice versa*. We apologize for the lack of clarity in our previous manuscript. We have removed the incorrect statement.

Comment 9: "Although these algae have relatively few genes and a small genome, they inhabit very different environments and are not phylogenetically related. Thus, their genome minimizations were probably the result of separate evolutionary processes." The authors again totally ignore the elephant in the room in that the ancestral algal genome was very likely small; did the authors even check the distribution of genomes sizes in a phylogenetic framework?

Response: It is possible that the existing microalgal genomes has decreased in size or constantly small during evolution. We believe that it is extremely difficult to discuss the validity of these possibilities on the basis of our data. To ensure that we do not exclude either possibility, we have changed the relevant text to the following:

"It is unclear whether the common ancestor of these microalgae had a small or large genome. Either way, the inclusion of organisms with extremely small genomes in phylogenetically distant clades implies that maintaining small and simple cells may be a general survival strategy." (Line 272 - 275)

Comment 10: Lines 68-69: "Thus, the minimal microalgal genome may provide clues regarding the core gene set required by plants". Plants or land plants? *M. hakoo* is on a completely different branch of the Viridiplantae than *Arabidopsis* and looking at charophytes would be a lot more informative for that purpose. Sure *M. hakoo* may not require as many housekeeping genes but that knowledge will not translate to land plants. If the goal is to find the minimal set of genes for all plants (not land plants), then focusing on protists is the way to go. Throwing in *Arabidopsis* just leads to more confusion (+ *Physcomitrella patens* appears out of the blue in Figure 3 !?). More importantly, a core set of genes is defined as the minimum set required for an organism to function, so the authors are basically saying here that finding the core set of genes will help find the core set of genes!

Response: In this case, "plant" was used to refer to algae and land plants (i.e., Archaeplastida). We know that Viridiplantae is one of the subkingdoms of the plant kingdom, and that it is a large phylogenetic group comprising land plants and green algae. However, as you point out, we should use the term "plant" carefully to prevent readers from misinterpreting what we are trying to say. We believe that the information regarding *M. hakoo* is related to and provides important insights into the biology of land plants. Nevertheless, to address your comments, we have eliminated as much of the comparison with *Arabidopsis thaliana* as possible. You mention that we should focus

on protists, but we have already performed a comparative analysis involving protists, including *C. merolae* and *O. tauri*, to identify the minimal gene set required for the survival of Archaeplastida species. The data for *Physcomitrella patens* were removed from Figure 3b (now moved to Figure 4b) and the above-mentioned description was changed as follows:

“To predict the core genes conserved in Archaeplastida species, we analyzed the genome of a microalga with a simple cell structure.” (Line 65 - 66)

Comment 11: This dubious and/or confusing logic is pervasive throughout the manuscript. For example, on lines 226 -228: "Moreover, similar to *C. merolae*, *M. hakoo* lacks plant-specific NSE5 and NSE6 orthologs (Supplementary Table 5), which are the non canonical and evolutionarily non-conserved components of the SMC5/6 complex. In *A. thaliana*...". Plant specific as in land plants? If so, why would anyone expect them in algae? If they are non-canonical, why expect them at all? Of course more complex organisms will have evolved more molecular complexity over time. On lines 232-232: "Accordingly, it is possible that SMC5/6 is more functionally limited in *M. hakoo* than in flowering plants." Again, why compare to land plants?

Response: According to your suggestion,, we have removed the comparative analysis and discussion involving land plants.

Comment 12: Overall, the manuscript is very verbose yet contains very little information at times. For example, on lines 217-220 "Our analysis confirmed that canonical components of all SMC complexes are conserved in *M. hakoo* (Supplementary Table 4), suggesting that SMC complexes are indispensable for the chromosome maintenance in *M. hakoo* and in other plant species." A basic set of SMC proteins is essential in eukaryotes. No chromosome maintenance? Death. This sentence is not informative and setups the meaningless SMC5/6 comparison with Arabidopsis. Again, on lines 250-253 "Therefore, our results suggest that mid-SUN is a fundamental INM protein and the acquisition of diverse factors enabled the development of relatively sophisticated and complex chromatin structures in phanerophytes." More complex organisms, more complex machinery? How is that informative? Why is that even in the RESULTS section. This is not a result.

Response: Although we think the fact that even green algae with small genomes have SMC proteins is useful information, we have deleted the statement about SMC conservation as well as Supplementary Table 4. Additionally, as suggested, we deleted the discussion about mid-SUN.

Comment 13: Other examples of confusion: table 2 Comparison of the genome data for microalgae => *Arabidopsis* is not an alga...

Response: The data for *Arabidopsis thaliana* were removed from Table 2.

Comment 14: Lines 178 "In summary, we revealed that *M. hakoo* has one of the smallest genomes among Viridiplantae species, and its gene sets that regulate the central dogma..." What dogma? This makes no sense.

Response: As suggested, we have revised the indicated statement as follows:
"In summary, we revealed that *M. hakoo* has one of the smallest genomes among Viridiplantae species, and its gene sets that regulate transcription and RNA transport are relatively simple compared with those in other algal species." (Line 218 - 221)

Comment 15: Lines 315-318 "Thus, we speculated that a high frequency of the G4 consensus sequence is a characteristic of the *M. hakoo* genome, and G4-related biological processes may have contributed to the increase in the genomic G+C content in this species during evolution." Why would telomere repeat units influence the full GC content of an organism? There is no basis for this speculation, this is not a result and certainly should not be written in the RESULTS section, and there is no explanation/hypothesis whatsoever about any mechanism that would explain this speculation.

Response: The G4 structure is present in telomeres as well as within chromosomes. Moreover, it influences transcription and replication. In fact, the G4 consensus sequence is present throughout the *M. hakoo* genome and it may affect the G+C content. We have revised the manuscript to discuss potential effects of G4 on the genomic properties in the Discussion section.

Reviewers' comments:

Reviewer #1 (Remarks to the Author):

I am glad the editor kept me in the loop for this review. I do think that the underlying data is valuable to publish. It does seem that the manuscript is telling a story that is not well-supported by the data or at least the current interpretation of the data. I consider the claims to find the most minimal algal genome to be mostly unsubstantiated, and the authors' interpretations of their findings to lack logic or perhaps subject knowledge of the underlying biological processes.

I was also disappointed by their latest responses to the other reviewer. As a superficial example, the answer to Question 2 completely missed the point. The reviewer stated : "6932 out of the 7628 (90.1%) of the M. hakoo proteins are annotated as hypothetical proteins in the provided file" to which the authors replied "approximately 80.7% of the CDSs have been annotated ..." This must be a language translation error because the reviewer was not questioning whether they had been annotated but rather why most of their annotations were hypothetical proteins.

One major flaw that is now apparent (perhaps with the rewriting of the manuscript) is that they attempt to define a core set of genes for Archaeplastida by surveying a minute fraction of available sequenced species. This is absurd. Their sample size is way too small. And is, in most cases, not even provided for the reader.

The authors also need to correct English throughout the manuscript for legibility.

Point-by-point response letter

Comment 1: I am glad the editor kept me in the loop for this review. I do think that the underlying data is valuable to publish.

Response: Thank you for your positive evaluation of our current data for *Medakamo hakoo*.

Comment 2: It does seem that the manuscript is telling a story that is not well-supported by the data or at least the current interpretation of the data. I consider the claims to find the most minimal algal genome to be mostly unsubstantiated, and the authors' interpretations of their findings to lack logic or perhaps subject knowledge of the underlying biological processes.

Response: We appreciate your comment and agree that the idea of finding the minimal genome is beyond the scope of this study. In the current version, we have revised the text and structure of the manuscript to focus primarily on the properties and genomic characteristics of *Medakamo hakoo*. We revised the title from “Genomic analysis of an ultrasmall green alga to predict the minimal plant gene set” to “Genomic analysis of an ultrasmall freshwater green alga, *Medakamo hakoo*”. We also deleted the description of the minimal genome from the Introduction. In the newly revised Figure 5, we present data on our analysis of gene sets common to 15 microalgal species.

Comment 3: I was also disappointed by their latest responses to the other reviewer. As a superficial example, the answer to Question 2 completely missed the point. The reviewer stated : “6932 out of the 7628 (90.1%) of the *M. hakoo* proteins are annotated as hypothetical proteins in the provided file” to which the authors replied “approximately 80.7% of the CDSs have been annotated ...” This must be a language translation error because the reviewer was not questioning whether they had been annotated but rather why most of their annotations were hypothetical proteins.

Response: We sincerely apologize for any misunderstanding and misinterpretation of the comment by the reviewer. After reassessment, we found that the software that we previously used for gene annotation, the “funannotate pipeline”, tended to annotate proteins as ‘hypothetical’ even when the expected protein function information is available. Therefore, we used the eggNOG-mapper tool and reanalyzed the identified genome sequences. This resulted in improved annotation of 4704 genes in predicted open reading frames of the *Medakamo hakoo* genome, as described in Supplementary Data 1. In addition, we performed a downstream analysis with GhostKOALA, as summarized in Supplementary Data 2. We emphasize that the results of the reanalysis do not affect our claims and conclusion in the manuscript.

Comment 4: One major flaw that is now apparent (perhaps with the rewriting of the manuscript) is that they attempt to define a core set of genes for Archaeplastida by

surveying a minute fraction of available sequenced species. This is absurd. Their sample size is way too small. And is, in most cases, not even provided for the reader.

Response: We appreciate your valuable comments. We agree that the sample size for the comparative analysis was small in the previous manuscript and that defining a core set of Archaeplastida genes is a technically and conceptually difficult task, which is beyond the scope of this study. To address these issues, we ran a larger-scale comparative genomic analysis using 15 microalgal genomes, including *Medakamo hakoo*, and instead described the set of common genes identified using the OrthoFinder program. Accordingly, we thoroughly revised Figure 5.

Comment 5: The authors also need to correct English throughout the manuscript for legibility.

Response: We have once again carefully revised the text and structure of the manuscript. In addition, the revised manuscript has been checked by a professional language-editing service.

REVIEWERS' COMMENTS:

Reviewer #1 (Remarks to the Author):

I am satisfied with the revisions. This contribution to the algal biology field will be valuable.